behaviour/cognition/evolution

sexual selection, third-party punishment, third-party compensation, cooperation, social selection

**Author for correspondence:**
Eamonn Ferguson
e-mail: eamonn.ferguson@nottingham.ac.uk

# To help or punish in the face of unfairness: men and women prefer mutually-beneficial strategies over punishment in a sexual selection context

Eamonn Ferguson, Erin Quigley, Georgia Powell, Liam Stewart, Freya Harrison and Holly Tallentire

School of Psychology, University of Nottingham, Nottingham NG7 2RD, UK

(iD) EF, 0000-0002-7678-1451

Consistent with a sexual selection account of cooperation, based on female choice, men, in romantic contexts, in general display mutually-beneficial behaviour and women choose men who do so. This evidence is based on a *two-choice-architecture* (cooperate or not). Here we extend this to include punishment options using a *four-choice-architecture* ('punishing a transgressor', 'compensating a victim', 'both punishing and compensating' or 'doing nothing'). Both compensation (a self-serving mutually-beneficial behaviour) and self-serving punishment, are associated with positive mate qualities. We test which is preferred by males and chosen by female undergraduates. We further explore effects of trait empathy and political ideology on these preferences. In a series of three studies using a third-party punishment and compensation (3PPC) game we show (Study One), that romantically-primed undergraduate males, express a preference to either 'compensate' or 'both compensate and punish', and undergraduate women find males who 'compensate' or 'compensate and punish' the most attractive (Studies Two and Three). Compensating men are perceived as compassionate, fair and strong by undergraduate women (Study Three). High trait empathy (Studies One and Three) and a left-wing political ideology (Study Three) are associated with a preference for compensation. Thus, self-serving mutually-beneficial behaviour can be preferred over self-serving punishment as a signal of mate quality in undergraduates. Implications for the evolution of cooperation are discussed with respect to sexual selection.

# 1. Introduction

Cooperation is a behaviour widely observed across all human societies [1] and constitutes a behaviour performed by one individual (actor) to the benefit of another (recipient) [2–4]. Explaining high levels of cooperation is problematic from an evolutionary perspective as it is not clear why someone would perform a behaviour that benefits another [3,5]. A number of ultimate ('why') explanations (e.g. kin selection, inclusive fitness) have been proposed to address this problem [2,6,7] and here we focus on another potential ultimate explanation: sexual selection [8,9]. Sexual selection describes the process of competition for access to mates that can lead to a number of long-term fitness advantages including increased: (1) fecundity, (2) generic quality of offspring, (3) parental care/resources and (4) chances that the offspring will survive to ultimately reproduce [8,10]. Sexual selection has led to the development of bodily ornaments, sexual dimorphisms and costly displays that enhance the actor's success when competing for a mate [8,10–13]. Displays of cooperation have been proposed as one such costly signal as they signify potentially desirable qualities in a mate of kindness, compassion and caring [11–15]. While sexual selection is a complex phenomenon encompassing female choice [10], male choice [8,10], mutual choice [16], cooperation between males to attract females [17], competition between females [18,19], moderation by genetic architecture [20] and the use of extended phenotypes [9] (see [8] for a recent review), we focus on female choice [10]. To date, the literature concerning the sexual selection of cooperation in humans has focused on female choice showing that women express a preference for men who cooperate [21,22]. It is to this literature we wish to add. This literature has only examined cooperation in terms of mutually-beneficial helping behaviour ([2–4,23]; see below) and not considered other proximal processes that support cooperation such as punishment [2,3]. As such, we examine: (1) the correspondence between male preference and female choice when punishment is an option as well as cooperation, and (2) the roles of other more proximal mechanisms such as emotions, empathy and political ideology.

## 1.1. Cooperation and sexual selection

Cooperation, while appearing on the surface a simple behaviour, whereby one individual (actor) benefits another (recipient) [3,4], it is more complex and requires, not only an understanding of the direct and indirect benefits to the actor and recipient(s), but also the immediate and long-term effects on the actor and recipient [2–4,6,7,24]. We use the semantic frameworks and typologies developed by West and colleagues [2], Bshary & Bergmüller [24] and Pizzari & Gardner [4] to define exactly what we mean by cooperation in our studies and the previous studies on the sexual selection of cooperation. In all these frameworks (+) represents an outcome that is beneficial and (−) an outcome that is a cost to either the actor and/or the recipient, with the first term always referring to the actor. Bshary & Bergmüller [24] further distinguish between *lifetime* and *immediate* fitness consequences, as well as the behaviour of the recipient as either *passive* or *dynamically interacting* with the actor, such that the recipient and actor *influence each other's* outcomes.

### 1.1.1. Previous studies on the selection of cooperation—a focus on mutually-beneficial cooperation

Previous studies have either used economic games (e.g. [23]) to examine male's preferences to help or not, or presented vignettes of helpful or unhelpful males (e.g. [21]) with women asked to make desirability judgements based on these vignettes.

Applying Bshary & Bergmüller's [24] framework to the studies that have used economic games, shows that a number of these have assessed what they term *self-serving mutually-beneficial behaviour*, which is defined as (+/+) with respect to *immediate* fitness outcomes with a *passive* recipient. Here both the actor and recipient benefit, even though the actor pays a cost. While the actor bears a small immediate financial cost, the actor also may gain short-term benefits in terms of warm-glow [25], as well as longer-term benefits in terms of: (1) increased reputation as a kind person which, via downstream indirect reciprocity [26], will increase their chances of being helped by others [27,28], (2) being chosen by others for mutual gain [29] and (3) being chosen as a mate [11,12]. These studies have involved charity dictator games [23], standard dictator games [30], online giving [31] and self-reported displays of wealth and generosity [32]. Others studies have explored what Bshary & Bergmüller [24] term *mutual cooperation* defined as (+/+) with respect to *immediate* fitness outcomes where the actor and recipient can *influence each other's* outcomes. These studies have involved public goods games [33–35] and the Prisoner's Dilemma [30]. Similar studies have examined female's ratings of a male's

desirability as a mate when presented with a vignette of a male either displaying a *self-serving mutually-beneficial behaviour* (e.g. a male buys food and drink for a homeless person, a male donates a £3000 work bonus to charity) or not (e.g. a male pretends to use a mobile phone and walks past a homeless person, a male spends a £3000 work bonus on a watch) [21,22].

Across these studies the picture is generally consistent with predictions from sexual selection theory based on female choice, with men altering their behaviour to choose the *self-serving mutually-beneficial behaviour* and/or *mutual cooperation* when in a romantic context and women preferring men who display *self-serving mutually-beneficial behaviour*. Three issues arise from this overview of the current literature. First, while the evidence shows that men choose *self-serving mutually-beneficial behaviour* and/or *mutual cooperation* when in a romantic mind-set and women also prefer *self-serving mutually-beneficial behaviour*, studies on women's choices have been separate from those on male preferences, and, to date, no study has explored this congruency by presenting men and women with the same cooperative task. Second, other proximal predictors of cooperation such as empathy and emotional responses have not been explored. Third, to date, the evidence for the sexual selection of cooperation, based on female choice, has focused only on *mutually-beneficial behaviour/cooperation*. This evidence has been based on experiments using a *two-choice-architecture* where participants can choose to either cooperate in some mutually-beneficial way or not [8–16,21,22] or women can express a preference for a male who has displayed some mutually-beneficial behaviour or not [21–22]. However, other processes that proximally support cooperation have not been explored within the context of the sexual selection of cooperation. In this paper we explore one of these—*punishment*.

### 1.1.2. The need to include punishment

The focus in the literature on the sexual selection of cooperation, by female choice, on mutually-beneficial behaviour, without the option to punish, is plausible as there are many cases where punishment is not possible. For example, many charitable appeals focus on helping victims of *natural disasters*, *war* or *illness* where there is no punishable transgressor [36]. However, in other cases, not only is a mutually beneficial response possible [37–41], but also the option to punish a transgressor (e.g. seeing a friend treated unfairly). Thus, we extend the study of the sexual selection of cooperation to include punishment as well as *self-serving mutually-beneficial behaviour*.

Punishment, whereby people pay a cost to punish a transgressor, has been shown to sustain high levels of cooperation [3,6,7,42–46]. In terms of Bshary & Bergmüller's [24] framework, punishment (−/−) is defined with respect to immediate fitness outcomes with a passive recipient. Indeed, increased levels of cooperation in the presence of punishment have been observed when punishment is implemented as either *second-party punishment* (2PP: direct punishment of a transgressor by another player in a repeated public goods game for example, at a cost to the punisher) or *third-party punishment* (3PP: punishment of a transgressor by an external agent, at a cost to that punisher) [47]. The punisher and recipient both pay an immediate cost (loss of finances) and the punisher a potential long-term cost in terms of loss of reputation due to being feared [48]. However, the punisher's behaviour may also signal desirable qualities in the punisher. For example, third-party punishers are perceived as trustworthy [49], with a desire to enforce norms of fairness [42,50,51], and have the strength to stand up to a transgressor [48,52]. Punishing a transgressor also has potential indirect benefits to the actor (self-protective) or the recipient (other-protective), in that it may deter the transgressor, and other potential transgressors, from future transgressions [53]. Therefore, the punishing behaviour can be defined as (+ & −/−). As there is a benefit to the punisher, it has been recommended that this is termed *self-serving punishment* [2].

Therefore, we move beyond the *two-choice-architecture* that has been used to study the sexual selection of cooperation and extend this to a *four-choice-architecture* that allows a third party to choose to either punish a transgressor, compensate a victim, both compensate and punish, or do nothing. This *four-choice-architecture* is studied widely in the context of restorative justice [39,54] and empirically examined using a *third-party punishment and compensation* (3PPC) game. The 3PPC game combines a third-party compensation (3PC) game (i.e. the actor, an external agent, can compensate a recipient who has been treated unfairly by another), with a 3PP game into a single set of choices and this allows us to model both self-serving mutually-beneficial behaviour and self-serving punishment behaviours in one set of choices. We use a 3PPC game to explore men's choice preference and explore if women's preference is congruent with men's displays in a 3PPC game. We further explore the role of empathic traits and political ideology as additional proximal influences on these preferences.

## 1.2. 3PPC game: mutual-benefit, self-serving punishment, free-riding and sexual selection

Table 1 shows how the 3PPC game choices (punish, compensate, both compensate and punish, or do nothing) map onto the frameworks of Bshary & Bergmüller [24] and West and colleagues [2–4].

*Compensating a victim* (+/+), as well as having a *direct* other-regarding component (helping the recipient), allows the actor to display their positive qualities. While incurring a small financial short-term cost, the actor displays qualities of compassion and kindness [37–39] that are likely to lead to longer-term benefits to the actor by enhancing their reputation as a nice person, and via downstream indirect reciprocity [26], leading to the increased probability that they will be helped in the future by others [28]. They are also more likely to be chosen by others to cooperate in mutually beneficial tasks [29] and selected as a mate [11,12,22]. The actor may also gain some personal utility in terms of feelings of warm-glow associated with helping [25]. The recipient also benefits directly in terms of both financial compensation and positive emotional outcomes. For example, the compensated victim may feel gratitude [55,57] that motivates them to help others via upstream indirect reciprocity [56]. Thus, following Bshary & Bergmüller [24] we refer to this as *self-serving mutually-beneficial behaviour*.

*Punishment* by a third-party punisher, while incurring a small immediate financial cost to the punisher, also signals desirable qualities of the punisher. There may also be a potential long-term cost if the punisher is feared [48] in terms of associated perceptions of violence [58]. Punishers, however, are also perceived as trustworthy [49], with a desire to enforce norms of fairness [42,50,51], and having the strength to stand up to a transgressor [48,52]. Punishing a transgressor also has potential *indirect* benefits to the actor (self-protective) or the recipient (other-protective), in that it may deter the transgressor, and other potential transgressors, from future transgressions [53]. As there is a benefit to the punisher it has been recommended that this is termed *self-serving punishment* [2].

*Both compensating and punishment* should entail the mutual benefits of compensation and the self-serving benefits of punishment, thus we term it *mutually-beneficial helping and punishment* and is akin to strong reciprocity.

*Doing nothing*, while incurring no financial cost to the actor, may incur a reputational cost to the actor, as this choice potentially signals lack of compassion towards the victim [27,28]. As this behaviour is purely self-serving we term it *free-riding* [2,59].

How will this *four-choice-architecture* play out in a sexual selection context? Altering choice architectures alters people's choice preferences [60,61]. When people can choose between punishment and compensation, people are more likely to choose to compensate [62,63], with punishment rates lower than when people play separate 3PC and 3PP games [38]. How might this be further affected in a sexual selection context? In a simple *two-choice-architecture* of mutually-beneficial helping or not, positive qualities are signalled by helping [64]. In a *two-choice-architecture* with self-serving punishment or not, punishment signals trustworthiness, for example [49]. In a sexual selection context, where self-serving mutually-beneficial behaviour and self-serving punishment each signal very different, yet desirable qualities [49], it is less clear which choice signals the potential mate's best qualities. Thus, two questions remain, at present, unanswered, for the sexual selection of cooperation based on female choice. How would men choose to display their qualities as a mate if they could both compensate and/or punish? How would women view male attractiveness based on men's decision, following a transgression, when both options are available?

## 1.3. Ratings of mate: beyond attractiveness

Desirability ratings have been used primarily to explore women's preferences for cooperative males [21,22,65,66]. However, Miller & Todd [12] propose a lens-model indicating that women select men based on a *set* of key characteristics that represent different phenotypic qualities of a potential mate. These include overall attractiveness, as well as signals of kindness and generosity linked to the development of cooperative relationships, and 'social status' to indicate resource holding. Thus, we extend the previous work to assess attractiveness, compassion, fairness and strength. As well as being theoretically consistent with Miller & Todd's [12] model of mate attraction, compassion and fairness are also major components of proximal mechanisms of cooperation such as reciprocity (both direct and indirect), inequality aversion [67,68] and empathy [64]. A further assumption of the lens-model is that all these markers load on a single latent factor of overall *mate quality*.

**Table 1.** Classification of behaviours in the 3PC game.

| preference | actor | recipient | term |
|---|---|---|---|
| compensate a victim | (−) *short-term costs* with respect to loss of financial/material resources<br>(+) *longer-term benefits* via 'downstream indirect reciprocity' resulting in an increased probability of being helped by others due to an enhanced reputation for kindness [26,54] and more likely to be chosen by others to cooperate in mutually-beneficial tasks [29]. *Short-term benefit* of increased feelings of warm-glow [25] | (−) *short-term costs* with respect to loss of financial/material resources<br>(+) *short-term benefit* in terms of direct other-regarding benefits of compensation of resources. *Longer-term benefits* resulting from feelings of gratitude potentially motivating helping of others via 'upstream indirect reciprocity' [54,55] | *self-serving–mutual benefit* [2–4,24] |
| punish the transgressor | (−) *short-term costs* with respect to loss of financial/material resources<br>(−) *long-term costs* with respect to a reputation for violence<br>(+) *longer-term benefits* of being perceived as trustworthy [48], with a desire to enforce norms of fairness [42,49,50], and have the strength to stand up to a transgressor [48,51]. Indirect self-protective (as well as other-protective) benefits in terms of deterring potential future transgressions [52] | (−) *short-term costs* with respect to loss of financial/material resources | *self-serving punishment* [2] |
| both compensate a victim and punish the transgressor | (−) *short-term costs* with respect to loss of financial/material resources<br>(+) *longer-term benefits* are the combined benefits from 'compensating a victim' and 'punishing the transgressor' | (−) *short-term costs* for the transgressor with respect to loss of financial/material resources<br>(+) *short- and longer-term benefits* are the benefits from 'compensating a victim' | *mutually-beneficial helping and punishment* |
| do nothing | (−) *longer-term costs* with respect to reputation loss as a nice/kind person and increased probability of endorsing transgression/unfairness as normative<br>(+) *short-term benefit* in terms of keeping financial/material resources | (−) *longer-term costs* for the victim in terms of reduced mood, feeling not cared for and unprotected. Therefore, less likely to engage in cooperative behaviour [54,55]<br>(+) *longer-term benefit* for the transgressor in terms of keeping financial/material resources and avoiding punishment | *free-riders* [2,56] |

## 1.4. Relationship length

While relationships can vary along many dimension (e.g. casual/serious), work on the sexual selection of cooperation has consistently explored the moderating effects of short-term versus long-term relationships [22]. The choice to use this particular distinction is driven by sexual strategy theory [69,70], which shows that men and women often look for very different characteristics when considering short- or long-term relationships [69]. Thus, the way women apprise a male's qualities as a mate should depend on whether she is considering a short-term or a long-term relationship. For example, evidence shows that women find kindness and compassion more desirable in long-term relationship [22,71]. Here we explore, for the first time, how women make judgements about not only men's attractiveness, but also their compassion, fairness and strength as a function of relationship length and the preferences made by a man within the *four-choice-architecture* of the 3PPC game. For example, are men who choose to compensate judged by women as attractive and also as compassionate, fair and strong, especially for long-term relationships.

## 1.5. Proximal processes: emotions, traits, ideology and correlated trait effects

While sexual selection may offer an ultimate ('why') explanation for cooperation, a number of proximal ('how') mechanisms are required to support it. We have suggested (table 1) that proximal mechanisms like reciprocity, warm-glow and norm enforcement may operate within the *four-choice-architecture* of the 3PPC game. However, other proximal mechanisms, such as emotional responses to the transgressor, are also likely to predict preferences. For example, people may feel moral outrage towards a transgressor, which is known to predict both a preference to punish the transgressor [52] and/or compensate the victim [72]. Thus, we explore if moral outrage is linked to a preference to punish, compensate or do both. Negative state relief theory suggests that people help a victim to manage their own negative emotions, in particular empathic distress and empathic sadness, that arise from witnessing the victim's distress [73]. As such, both empathic distress and sadness should be associated with a preference to compensate.

Empathy, both as an emotional response and as a trait, is a proximal mechanism, known to increase the likelihood of helping a victim [73–76]. As such, those with higher levels of trait empathy should display a stronger preference to compensate rather than punish [50,77]. An association between empathic traits and the potential to favour a compensatory preference further suggests that empathic traits may be sexually selected as a correlated trait [78]. That is, if men higher in empathy choose to compensate and women higher empathy prefer men who compensate, trait empathy becomes selected as a correlated trait of the preference to compensate [78].

Similarly, political ideology is also known to influence choices to punish a transgressor or compensate a victim [51]. Those with a right-wing conservative political ideology are more likely to hold 'small world' views [79] and prefer to punish transgressors [80], while those with a left-wing ideology and more politically liberal views prefer rehabilitation [51] and endorse fairness-based moral values [81]. Thus, we may expect those with a left-wing political ideology to prefer victim compensation and those with a right-wing ideology to punish a transgressor. The expressed preferences to punish or compensate may also signal underlying wider political preferences and this may attract those with similar ideologies. In this way political ideology may also be sexually selected as a correlated trait of a preference to punish or compensate [78]. Consistent with this is evidence that political ideology has a genetic component [82].

## 1.6. Current studies

We report three studies that all adopt the *four-choice-architecture* of a 3PPC game: (1) punish the transgressor, (2) compensate the victim, (3) do a mixture of punishment and compensation, or (4) do nothing [49,83]. Study One explores how heterosexual men's preferences in a 3PPC game vary as a function of being in a romantic context (primed by describing their ideal date [32]) or not and how these are further predicted by empathic traits and emotions that arise as a consequence of a transgressor's actions. Study Two uses a within subjects design to examine heterosexual women's ratings of men's attractiveness, for a short-term and a long-term relationship, as function of the males preference to: (1) punish, (2) compensate, (3) do a mixture of punishment and compensation, or (4) do nothing in a 3PPC game. Study Three uses a mixed (between-within) subjects design to examine heterosexual women's rating of men's attractiveness, compassion, fairness and strength, as a function

of males preference to: (1) punish, (2) compensate, (3) do a mixture of punishment and compensation, or (4) do nothing in a 3PPC game. As in Study Two, these ratings were made as a function of considering men for a short-term or a long-term relationship. We also examine how women's trait empathy and political ideologies affect their ratings.

# 2. Methods

The materials used in Studies One, Two and Three can be found in the electronic supplementary material, section A '*Materials for Studies 1, 2 and 3*'.

## 2.1. Study One

### 2.1.1. Sample

Eighty-three heterosexual males were assigned to either a 'romantic prime' ($N = 30$: mean age = 19.0, s.d. = 1.2, 70% single), 'pure-control' ($N = 30$: mean age = 19.6, s.d. = 1.1, 67% single) or an 'active control' for priming ($N = 23$: mean age = 19.5, s.d. = 1.5, 78% single).[1] Seventy-three participants (88.0%) reported their ethnicity as White British, three (3.6%) as Black British, two (2.4%) as White Irish and the remaining five as British Asian (1.2%), Chinese (1.2%), British India (1.2%), Arab (1.2%) and Indian (1.2%). There are no studies to guide a power calculation for a sexual selection study with a one-shot 3PPC game. The closest experiments have examined a single donation in a *two-choice-altruism-architecture* [23,33]. Based on these experiments the average effect size is large (Cohen's $d$ = 0.84). Based on this effect size estimate, with an $\alpha$ of 0.05, we need 23 participants per condition (romantically primed versus controls) to achieve a power of 0.80.

### 2.1.2. Primes

We used the priming procedures described in [32]. The 'romantic prime' involved participants viewing photographs of three women, selecting the one they thought was the most attractive and writing, for up to three minutes, a brief summary of their ideal date with her [32]. An 'active control' was included to ensure that any effects for the 'romantic prime' reflected the prime content (romantic) rather than being primed *per se*. This involved the participants choosing one of three ordinary street scenes in the UK and spending up to three minutes writing about most pleasant weather conditions in which to walk around and look at the buildings on that street. Finally, a 'pure-control' (no prime) was included to reflect the standard control condition in 3PPC games.

### 2.1.3. Third-party punishment and compensation (3PPC) game

Participants played a standard one-shot 3PPC game [39]. A one-shot game is used as evidence from the theory of Revealed Altruism suggests that initial allocation responses reflect the underlying cooperative preference that participant wishes to display [84–86]. Participants read the instructions to the game in *private* and were told that the game involved three people (A, B and C). Participants were presented with a schematic of the game and were informed that Person A has been *given* £10 and told that they can share some, none or all of it with player B. As this is unearned house-money a 50 : 50 share would be fair [87]. Player B has £0. Player B has to accept whatever Player A decides to give them. Player C has £5. Player C can choose to spend some, none or all of their £5 to either (1) *compensate player B*, (2) *punish player A*, (3) *do a mixture of compensation and punishment* or (4) *do nothing* and keep the money. The decision was made efficient as every £1 player C spends to compensate player B will increase B's earnings by £2, and every £1 player C spends to punish player A reduces A's earnings by £2. Once they had read the instructions, participants completed a set of comprehension questions. The experimenter returned and went through the answers. If all were correct, the participant continued to the actual game. If there were incorrect answers these were explained and participant continued to the

---

[1]Assignment to the 'pure-control' and 'romantic prime' was randomized. The 'active control' was added at the request of a reviewer five months later. We did not initially include an 'active control' as: (1) results reported in [12] showed the effect of a 'romantic prime' is due to its content rather than the process of priming *per se* and (2) in a 3PPC game a pure-control is the standard control. As the results for the 'active control' and the 'pure-control' are the same (see electronic supplementary material, section B, table S1 and figure S1) we combined these. Furthermore, the demographics (1) age ($F_{(2,82)} = 0.621$, $p = 0.540$) and (2) relationship status ($\chi^2_2 = 0.878$, $p = 0.654$) did not vary across the three conditions.

actual game once they understood the task. The actual game was played in private and decisions were made anonymously. Participants were told that they would be player C, and faced a scenario where player A had given £2 to player B. The participant then made a decision to: (1) punish player A, (2) compensate player B, (3) both punish and compensate or (4) to do nothing and keep the £5 for themselves. They could choose only one option. Participants were informed that the game between A and B was hypothetical but that they were playing for real money and the choice they made would constitute their final pay-off if they were selected to be paid. Given that players A and B were hypothetical then the decision made by the participant, player C, will affect their own pay-off but not players A and B. Thus, the participants decision signals their *intent* to punish or compensate. If participants were only concerned about the direct effects of their actions on players A and B, rather than what their actions signal about them, they would keep the money and do nothing. As there is evidence that the intentions are important signals about reputation [68,88] as well as self-signals about the person's own motivations [89], we feel that participant's choices will be an indication of their preferences. Furthermore, there is evidence that preferences in hypothetical games are similar to those in real games [90–92].

To avoid deception and reduce transaction costs in such studies either a subset of tasks within an experiment [93] or subset of participants can be selected for payment. The evidence suggests that selecting participants, rather than tasks, has no discernible effects on the pattern of results [94]. Therefore, as we use a one-shot 3PPC game we selected to pay a subset of participants chosen at random. Eight participants would be randomly selected and paid based on their decision.[2]

### 2.1.4. Psychometric assessments of empathic/compassionate mood and traits

Participants used a series of mood adjectives to indicate their feelings as a consequence of Player A's transgression. Six adjectives (e.g. annoyed) assessed *moral outrage* ($\alpha = 0.90$: $M = 9.96$, s.d. = 5.60) [95]. Four adjectives (e.g. compassionate) assessed *empathic-concern/compassion* ($\alpha = 0.73$: $M = 8.90$, s.d. = 3.97), three adjectives (e.g. uneasy) assessed *distress* ($\alpha = 0.78$: $M = 84.84$, s.d. = 2.62) and three adjectives (e.g. sad) assessed *sadness* ($\alpha = 0.77$: $M = 5.38$, s.d. = 3.05) [96]. *Trait Empathic Anger* was assessed using the 'trait empathic anger' scale, where participants rated how each of seven statements describe them (e.g. 'I get angry when a friend of mine is hurt by someone else') on a five-point scale (1 = does not describe me very well to 5 = does describe me very well) ($\alpha = 0.78$: $M = 23.26$, s.d. = 4.81) [97]. Trait empathy was assessed in terms of *Trait Empathic Concern and Perspective Taking* from the Interpersonal Reactivity Index (IRI) [98]. Participants rated how each of seven statements described their *empathic concern* (e.g. 'Sometimes I don't feel very sorry for other people when they are having problems'; $\alpha = 0.81$: $M = 23.50$, s.d. = 5.07) and how each of seven statements described their *perspective taking* (e.g. 'When I'm upset at someone, I usually try to "put myself in his shoes" for a while'; $\alpha = 0.76$: $M = 823.60$, s.d. = 4.69). These were responded to on a five-point scale (1 = does not describe me very well to 5 = does describe me very well).

### 2.1.5. Design and procedure

Participants played the 3PPC game individually, in private, in an experimental cubical. Participants initially completed the priming task (if in a priming condition) in private and placed their narratives in sealed envelopes in a sealed box to maintain anonymity. Analyses of men's free-response romantic narratives show they are primarily other orientated (electronic supplementary material, section D, table S5). After the priming phase, the experimenter returned and introduced participants to a one-shot 3PPC game [39]. Participants completed the actual 3PPC in private and decisions were made anonymously and placed in sealed envelopes by the participants, who then placed these envelopes in a sealed post-box. There was approximately 5 min between the end of the priming task and the start of the 3PPC game. On completion the experimenter returned and gave the participants the post study survey. Again, this was completed in private and placed in an envelope. Participants were then debriefed.

---

[2]Overall eight participants were paid. For the no-prime and romantic-primes conditions, which were run first, participants were told that five would be randomly selected. We later added the active-prime condition and in that we told participants that three would be selected at random.

## 2.2. Study Two

### 2.2.1. Sample

For comparability with Study One the analyses focused on 104 heterosexual women (mean age 26.3, s.d. = 11.5: 58% in a relationship: four lesbians, six bisexuals and five women who did not specify their sexuality were excluded after data collection). Farrelly *et al.* [21] report that cooperative males are perceived as more attractive for a long-term versus a short-term relationship and provide an effect size for the interaction of $r = 0.21$. Based on this effect size we would need 64 participants for a two by four within subjects design to identify an interaction with a power of 0.80 and an $\alpha$ of 0.05.

### 2.2.2. Design and procedure

The study was a two (*relationship type*: short-term versus long-term) by four (*preference*: punish, compensate, both or nothing) within subjects design.

As in previous studies we used vignettes to explore women's preferences [21]. Women were presented with a vignette describing the same 3PPC game as in Study One whereby women had to make ratings based on how men (player C) responded (either punishing player A, compensating player B, both punishing player A and compensating player B or keeping the money) to the unfair transfer from player A to B. In all cases women responded and made their choices individually. However, these individual choices were made in two contexts. Forty-four per cent of the women made their individual responses on their own with no-one else present and 56% made their individual responses with others present in a lecture setting. Initially they rated how unfair player A was (1 = Not at all, 7 = Completely). They were then asked to imagine person C was male and, if they were looking for a *short-term* relationship, to what extent would they find him attractive if he decided to spend his money to (1) punish player A, (2) compensate player B, (3) both punish and compensate or (4) keep the money. These ratings were made on a 7-point scale (1 = Not at all Attractive to 7 = Very Attractive). Ratings below 3.5 on this scale may be interpreted in terms of degrees of *unattractiveness* and greater than 3.5 in terms of degrees of *attractiveness* (electronic supplementary material, section C, table S2). They then made the same ratings when considering a *long-term* relationship.

## 2.3. Study Three

### 2.3.1. Sample

The analyses focuses on 161 heterosexual women (mean age 26.78, s.d. = 8.14: 57% in a relationship: five lesbians, 12 bisexuals and three women who did not specify their sexuality were excluded after data collection). Eighty heterosexual women were in the short-term condition and 81 in the long-term condition. The interaction term in Study Two has an effect size of $r = 0.24$ (Cohen's $d = 0.49$). For a two between (long-term versus short-term relationship) by four within (restorative justice preferences) design (assuming an association of $r = 0.50$ between measures) we need 66 participants per condition to achieve 0.80 power with an $\alpha$ of 0.05.

### 2.3.2. Design and procedure

This study was a two between (*relationship type*: short-term versus long-term) by four within (*preference*: punish, compensate, both or nothing) subjects design. Women were randomly allocated to *relationship type*.

As in Study Two, women were presented with a vignette describing the same 3PPC game as in Study One whereby women had to make ratings based on how men (player C) responded (either punishing player A, compensating player B, both punishing player A and compensating player B or keeping the money) to the unfair transfer from player A to B. There were three modifications to Study Two. First, to confirm that the results observed in Study Two are not due to order effects for relationship type, women were randomly allocated to consider either a short-term or long-term relationship. Second, we extended the theoretical breadth of ratings for Player C to include: (1) *compassion* (1 = Uncompassionate, 4 = Neither Compassionate nor Uncompassionate, 7 = Very Compassionate), (2) *fairness* (1 = Unfair, 4 = Neither Fair nor Unfair, 7 = Very Fair), and (3) *strength* (1 = Weak, 4 = Neither Weak nor Strong, 7 = Strong), as well as *attractiveness* (1 = Unattractive, 4 = Neither Attractive nor Unattractive, 7 = Very Attractive). Third, we include measures of women's trait empathic concern and political ideology.

### 2.3.3. Empathic concern and political ideology

We assessed women's trait empathic concern using a four-statement version of Davis's IRI empathic concern scale, with each statement scored on a seven-point scale [99] ($\alpha = 0.82$, $M = 21.72$, s.d. = 4.60) and a single item 11 point index of left- versus right-wing ideology (0 = left-wing and 10 right-wing: [100]: $M = 3.97$, s.d. = 1.97).

## 2.4. Cultural context of the studies

In terms of the cultural context of the experiments, all participants were sampled from Nottingham University in the UK (electronic supplementary material, section A, for demographics for Nottingham and Nottingham University).

# 3. Results

## 3.1. Study One

### 3.1.1. Pattern of preference as a function of primes

The pattern of preferences did not significantly vary between the 'pure-control' and 'active control' (see electronic supplementary material, section B, table S1 and figure S1). However, there were significant differences for expressed preferences between the 'romantic prime' and both the 'pure-control' and the 'active control' conditions (see electronic supplementary material, section B, table S1 and figure S1). These differences were virtually identical with men exposed to either of the controls ('pure' or 'active') more likely to express a preference to 'do nothing' or 'punish', and men exposed to the 'romantic prime' more likely to express a preference to 'compensate'. Thus, we combined the 'pure-control' and 'active control' into a single 'combined control' group. The distribution of preferences by 'combined control' versus the 'romantic prime' are shown in figure 1. Yate's corrected Chi-squares, to account for some small cell Ns (there were no expected cells < 5), with $\varphi$ as an index of effect size (0.2 = small, 0.3 = medium and ≥0.5 = large) showed that that men exposed to the 'combined control', compared with the 'romantic prime', were more likely to express a preference to 'do nothing' ($\chi^2_{\text{yates corrected}} = (1)\ 12.53$, $p = 0.000$: $\varphi = 0.415$) or 'punish' ($\chi^2_{\text{yates corrected}} = (1)\ 6.914$, $p = 0.015\ \varphi = 0.320$). Conversely, men exposed to the 'romantic prime', compared with the 'combined control', were more likely to express a preference to either 'compensate' ($\chi^2_{\text{yates corrected}} = (1)\ 15.552$, $p = 0.001$: $\varphi = 0.461$) or 'both compensate and punish' ($\chi^2_{\text{yates corrected}} = (1)\ 5.86$, $p = 0.024$: $\varphi = 0.298$). The dominant financial distributions for the preference to 'both compensate and punish' was either to (1) equally spend on both or (2) spend more to compensate than punish (electronic supplementary material, section D, table S3). Controlling for participants' relationship *status* did not alter these results (electronic supplementary material, section D, table S4). Given the effect sizes between the 'romantic prime' and 'combined control' conditions, 19 participants are needed to achieve a power of 0.80 with an $\alpha$ of 0.05 for variation in displays of compensation, 19 for 'doing nothing', 30 for punishment and 48 for 'both compensate and punish'.

### 3.1.2. Role of empathic traits and emotion

Zero-order associations between the traits and emotions can be found in electronic supplementary material, section D, table S6. We ran a series of multi-nominal regression models (table 2) to explore the effects of traits and emotions on choice of preference. Those who expressed a preference to 'do nothing' acted as the reference category in all models.

Model 1 contained only the experimental conditions and the results indicated that compared with a preference to 'do nothing' those who expressed a preference to either 'compensate' or 'compensate and punish' were more likely to have been exposed to the romantic prime. Models 2–4 added traits and emotion parameters to the effects of the experimental conditions: Model 2 (traits added), Model 3 (emotions added) and Model 4 (both traits and emotions added). The effect of the romantic prime on the preference to 'compensate' or 'compensate and punish' remained significant across Models 2–4. Higher levels of trait empathic concern predicted a preference to 'compensate' but this effect dropped out of the model when emotions were added. However, higher levels of trait 'empathic concern' and trait 'perspective taking' predicted a preference to 'both compensate and punish' and these remained

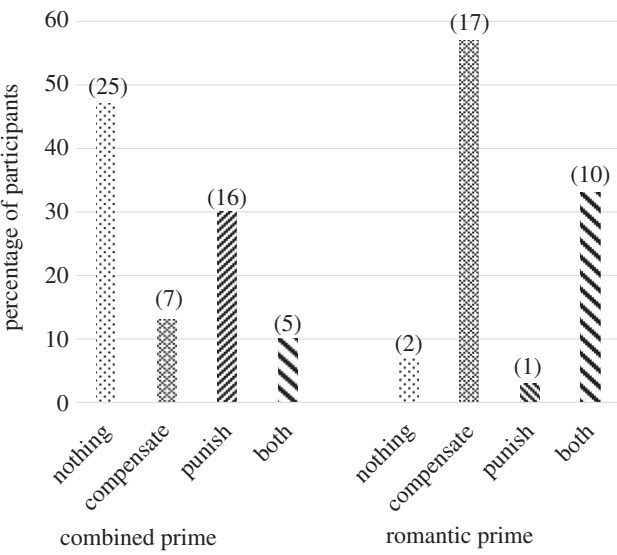

**Figure 1.** Prime by preference. (Values in parentheses are the number of participants.)

significant across models. Emotions linked to the transgressor (Player A) did not significantly predict preferences. Thus, exposure to a romantic prime increases preference to either 'compensate' or 'both compensate and punish', and traits linked to compassion (empathic concern) and empathy (perspective taking) consistently predicted a preference to 'both compensate and punish' and trait 'empathic concern' predicting a preference to compensate in the trait-only model (Model 2).

## 3.2. Study Two

### 3.2.1. Perceived fairness of the transgression

Compared with a rating of 1 (completely fair), women rated Player A as unfair ($M = 4.70$, s.d. $= 1.50$; $t(103)_{\text{one sample}} = 25.17$, $p = 0.000$: Cohen's $d = 2.47$).

### 3.2.2. Attractiveness as a function of preference and relationship length

Figure 2 displays women's mean attractiveness ratings for men across preferences for both short- and long-term relationships.

A two (*relationship type*: short-term versus long-term) by four (*preference*: punish, compensate, both or nothing) repeated measures ANOVA was specified. Mauchly's test of Sphericity was significant for *preference* and the interaction between *preference* and *relationship type* (both $p$s $= 0.000$ and Epsilons greater than 0.80). As Sphericity cannot be assured, with an Epsilon greater than 0.75, Huynh–Feldt corrections were applied. There was a significant main effect for *relationship type* ($F_{\text{Huynh–Feldt }(1,268.79)} = 8.06$, $p = 0.005$, $\varepsilon^2 p = 0.073$), with overall attractiveness ratings higher for a short-term relationship 3.72 (95% CI 3.54, 3.90) versus 3.58 (95% CI 3.41, 3.75), a significant main effect for *preference* ($F_{\text{Huynh–Feldt }(3,268.79)} = 119.05$, $p = 0.000$, $\varepsilon^2 p = 0.539$), with overall attractiveness ratings significantly higher for 'compensation' (5.57 (95% CI 5.35, 5.80)) followed by 'doing both' (3.96 (95% CI 3.62, 4.29)), 'doing nothing' (2.58 (95% CI 2.29, 2.87)) and lastly 'punishment' (2.50 (95% CI 2.23, 2.77)). All means are significantly different from each other apart from 'doing nothing' and 'punishment'. There was also a significant interaction ($F_{\text{Huynh–Feldt }(3,268.79)} = 6.45$, $p = 0.000$, $\varepsilon^2 p = 0.059$), showing that women find punishment less *unattractive* for short- compared with long-term relationships ($t_{(102)} = 4.39$, $p = 0.000$: Cohen's $d = 0.31$). Women's current relationship *status* (single or not) or study *setting* (individual or group) did not alter the results (electronic supplementary material, section D, table S7).

## 3.3. Study Three

### 3.3.1. Perceived fairness of the transgression

Consistent with Study Two, compared with a rating of 1 (completely fair), women rated Player A as unfair ($M = 4.06$, s.d. $= 1.80$; $t(156)_{\text{one sample}} = 21.45$, $p = 0.000$: Cohen's $d = 1.71$).

**Table 2.** Preferences as a function of priming condition, affect, traits and mood. Coefficients are unstandardized; Condition = combined control (0) versus romantic prime (1); reference group = do nothing.

| | model 1 | model 2 | model 3 | model 4 |
|---|---|---|---|---|
| **punish** | | | | |
| condition | −0.25, p = 0.845 | 0.75, p = 0.599 | −0.54, p = 684 | 0.45, p = 0.782 |
| traits empathic concern | | 0.07, p = 0.428 | | 0.16, p = 0.160 |
| trait perspective taking | | 0.07, p = 0.347 | | 0.13, p = 0.150 |
| trait empathic anger | | 0.12, 0.146 | | 0.10, p = 303 |
| moral outrage (player A) | | | 0.15, p = 0.194 | 0.18, p = 0.132 |
| empathic concern (player A) | | | −0.14, p = 0.305 | −0.31, p = 0.058 |
| empathic distress (player A) | | | −0.18, p = 0.496 | −0.28, p = 0.341 |
| sadness (player A) | | | 0.23, p = 0.250 | 0.25, p = 0.210 |
| **compensate** | | | | |
| condition | **3.41, p = 0.000** | **4.63, p = 0.000** | **2.62, p = 0.000** | **4.69, p = 0.000** |
| traits empathic concern | | **0.20, p = 0.045** | | 0.17, p = 0.146 |
| trait perspective taking | | −0.002, p = 0.963 | | 0.001, p = 0.993 |
| trait empathic anger | | 0.06, p = 0.456 | | 0.08, p = 0.399 |
| moral outrage (player A) | | | −0.09, p = 0.495 | −0.08, p = 0.558 |
| empathic concern (player A) | | | 0.08, p = 0.510 | −0.008, p = 0.992 |
| empathic distress (player A) | | | 0.15, p = 0.570 | 0.01, p = 0.963 |
| sadness (player A) | | | 0.24, p = 0.256 | 0.26, p = 0.226 |
| **compensate and punish** | | | | |
| condition | **3.22, p = 0.001** | **4.71, p = 0.000** | **3.54, p = 0.000** | **4.66, p = 0.001** |
| traits empathic concern | | **0.34, p = 0.002** | | **0.28, p = 0.027** |
| trait perspective taking | | −0.02, p = 0.794 | | −0.04, p = 0.661 |
| trait empathic anger | | −0.08, p = 0.376 | | −0.08, p = 0.431 |
| moral outrage (player A) | | | −0.0, p = 0.557 | −0.02, p = 0.883 |
| empathic concern (player A) | | | 0.171, p = 0.182 | 0.06, p = 0.673 |
| empathic distress (player A) | | | 0.30, p = 0.275 | 0.08, p = 0.803 |
| sadness (player A) | | | 0.09, p = 0.695 | 0.15, p = 0.527 |
| AIC | 31.36 | 195.88 | 193.34 | 205.43 |
| $R^2$ (Cox and Snell) | 0.372 | 0.512 | 0.498 | 0.592 |

### 3.3.2. Fairness, empathic concern and political ideology

Player A was rated as being more fair by women who expressed a right-wing ideology ($\rho = -0.27$, $p = 0.001$) or had lower empathic concern ($\rho = 0.37$, $p = 0.000$). A greater left-wing ideology was associated with greater empathic concern ($\rho = -0.26$, $p = 0.002$).

### 3.3.3. Ratings as a function of preference and relationship length

Based on a final $N$ of 159, a two between (*relationship type*: short-term versus long-term) by four within (*preference*: punish, compensate, both or nothing) subjects, multivariate mixed MANOVA was conducted on the four ratings. Mauchly's tests of Sphericity were significant for *preference* for each of the ratings (attraction, compassion, fairness and strength: all $p s = 0.000$ and all Epsilons greater than 0.75). As Sphericity cannot be assured, Epsilon-adjusted effects are reported for effects containing the within subject factor. There was a significant effect for *preference* ($F_{\text{Pillai's Trace } (12,1410)} = 33.14$, $p = 0.000$, $\varepsilon^2 p = 0.220$), but no significant *interaction* ($F_{\text{Pillai's Trace } (12,1410)} = 1.62$, $p = 0.080$, $\varepsilon^2 p = 0.014$). The *preference* effect was significant for all four ratings: (1) attractiveness ($F_{\text{Huynh–Feldt } (2.67,419.97)} = 130.72$, $p = 0.000$, $\varepsilon^2 p = 0.454$), (2) compassion ($F_{\text{Huynh–Feldt } (2.66,418.17)} = 218.05$, $p = 0.000$, $\varepsilon^2 p = 0.581$), (3) fairness ($F_{\text{Huynh–Feldt } (2.75,431.54)} = 96.42$, $p = 0.000$, $\varepsilon^2 p = 0.380$), and (4) strength ($F_{\text{Huynh–Feldt } (2.70,424.48)} = 40.25$, $p = 0.000$, $\varepsilon^2 p = 0.204$). Figure 3 shows that for all four ratings, a preference to compensate, compared with the other three preferences resulted in significantly higher ratings of attraction, compassion, fairness and strength. A preference to 'punish' or 'do nothing' was associated with ratings of the male being unattractive, uncompassionate, unfair and weak, and these were significantly lower than a preference to 'compensate' or 'both compensate and punish'. A preference to 'both compensate and punish' also resulted in ratings of attraction, compassion, fairness and strength that were significantly higher than for preferences to 'punish' or 'do nothing', but significantly lower than a preference to just 'compensate'.

In terms of *relationship* type there was a significant multivariate effect ($F_{\text{Pillai's Trace } (4,151)} = 4.72$, $p = 0.001$, $\varepsilon^2 p = 0.109$) with significant univariate effects for (1) attractiveness ($F_{(31,157)} = 8.47$, $p = 0.031$, $\varepsilon^2 p = 0.029$) and (2) strength ($F_{(31,157)} = 7.45$, $p = 0.049$, $\varepsilon^2 p = 0.024$). Such that attractiveness was rated higher for a short-term relationship ($3.72_{\text{short-term}}$ [95% CI 3.57, 3.87] versus $3.49_{\text{long-term}}$ [95% CI 3.34, 3.64]), replicating the results of Study Two, and strength rated as higher for a long-term relationship ($3.93_{\text{short-term}}$ [95% CI 3.77, 4.08] versus $4.14_{\text{long-term}}$ [95% CI 3.99, 4.30]). The interactions were not significant for any of the ratings.

Thus, regardless of relationship length, women find men the most attractive, compassionate, fair and strong if they choose to 'compensate' over all other preferences and unattractive, unfair, uncompassionate and weak when they either 'punish' or 'do nothing'. This replicates the findings in Study Two with respect to attractiveness and shows they generalize also to ratings of compassion, fairness and strength.

### 3.3.4. Covariance of ratings

Based on Miller & Todd's [12] lens-model we would expect that the four ratings (attractiveness, compassion, fairness and strength) to positively covary with each other to form a single latent factor of overall 'mate quality' (the summed combination of attractiveness, compassion, fairness and strength). However, context may matter and people's ratings of the target's overall 'mate quality' may take into account context in which the target is acting. In which case ratings of attractiveness, compassion, fairness and strength should all positively covary within each context (e.g. punish or compensate) (Model 1). However, there is also a possibility of *specific* overall ratings of the targets attractiveness, compassion, fairness and strength that are independent of context (compensation, punishment, doing both or doing nothing). That is, people's four ratings of compassion are all associated with each other regardless of the targets actions (e.g. punish or compensate). With same true of fairness, strength and attractiveness (Model 2).

To explore these possibilities we used confirmatory factor analysis (CFA) to specify these two models in MPlus 8.1 [101] using a weighted diagonal least squares (with means and variance adjustments) estimator. Model 1 specified four factors that represented ratings of compassion, strength, fairness and attractiveness within each preference. Thus, factor-1 represented these four ratings when men 'both compensate and punish', factor-2 when men 'do nothing', factor-3 when men 'compensate' and factor-4 when men 'punish'. Model 2 also had four factors with factor-1 comprising all four compassion ratings regardless of the preference, factor-2 all four strength ratings, factor-3 all four fairness ratings and factor-4 all four attractiveness ratings. Fit was assessed in terms of the Tucker–Lewis index (TLI, which should be greater

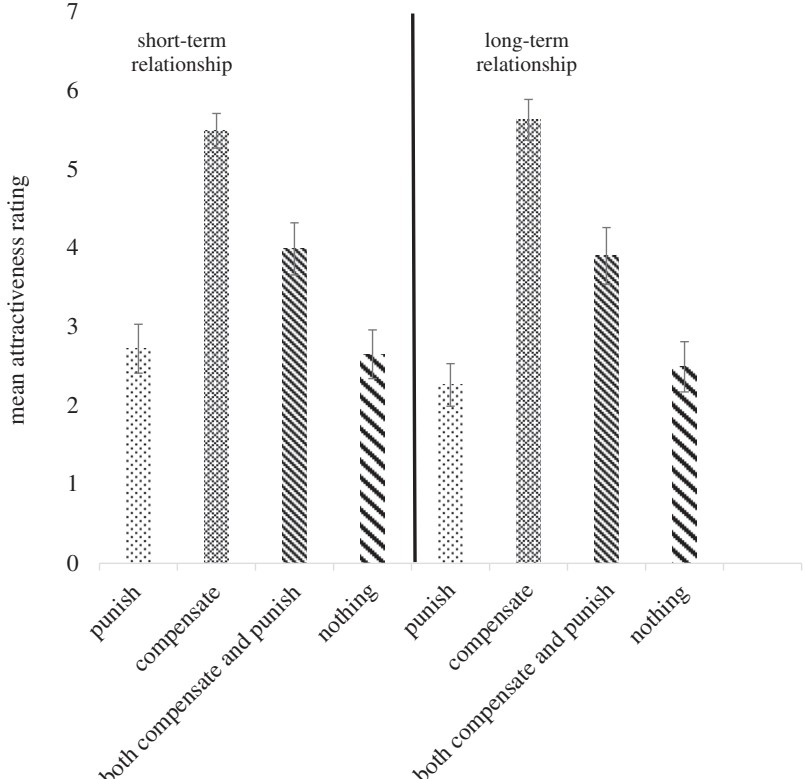

**Figure 2.** Mean attractiveness (error bars = 95% CIs) by preference and relationship length.

than 0.95), the comparatives fit index (CFI, which should be greater than 0.95) and the root mean square error of approximation (RMSEA, which should be 0.06 or less and non-significant) [102]. The fit for Model 1 was excellent (TLI = 0.98, CFI = 0.98, RMSEA = 0.06 RMSEA $p$-value = 0.112), whereas the fit for Model 2 was poor (TLI = 0.48, CFI = 0.58, RMSEA = 0.31 RMSEA $p$-value = 0.000). The factor loadings for Model 1 are shown in table 3. Therefore, when considering different preferences overall 'mate quality' is indicated by the sum of positive ratings of attractiveness, compassion, fairness and strength.

### 3.3.5. Effects for empathic concern and political ideology

To model the effects of the continuous trait of empathic concern and the continuous indicator of political ideology we ran a series of regression models using generalized estimating equations (GEE) ([103–105]; see electronic supplementary material, section E for modelling details).

Table 4 details the model for empathic concern. The significant effect for 'rating type' indicates that compared with an overall rating of strong ($M$ = 4.05 [95% CI = 3.93, 4.15]), fairness ($B$ = −0.26, $p$ = 0.013; 95% CI = −0.472, −0.056), compassion ($B$ = −0.81, $p$ = 0.000; 95% CI = −1.013, −0.601) and attractiveness ($B$ = −0.53, $p$ = 0.000; 95% CI = −0.755, −0.302) were all rated as significantly lower. The significant interaction of *rating type* by *preference* is depicted in figure 3.

There was a significant interaction between *empathic concern* and *preference*. This interaction describes how the strength of the association between empathic concern and the rating of a man's overall 'mate quality' (the summed combination of attractiveness, compassion, fairness and strength) varies as a function of each preference (compensate, punish or both) with 'do nothing' acting as the reference. Compared with the preference to 'do nothing', there was a significantly stronger positive association between levels of empathic concern and overall 'mate quality' for: (1) 'both compensate and punish' ($B$ = 0.103 $p$ = 0.001; 95% CI = 0.045, 0.162) and (2) 'compensate' ($B$ = 0.152 $p$ = 0.001; 95% CI = 0.094, 0.209). Thus, higher levels of empathic concern were predictive of higher ratings of overall 'mate quality' when the male chose to either 'both compensate and punish' or 'compensate' compared with 'doing nothing'. The association between empathic concern and overall 'mate quality' for the preference to 'punish' ($B$ = 0.044, $p$ = 0.118; 95% CI = −0.011, 0.100) was not significantly different from the preference to 'do nothing'. Adding in the additional

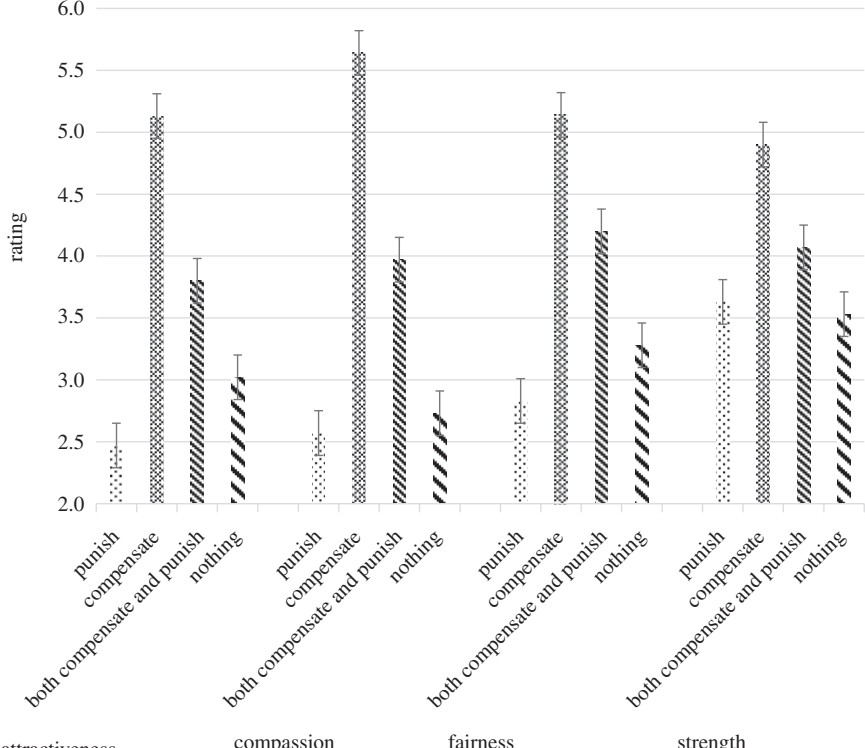

**Figure 3.** Women's ratings of men as a function of the man's preference (error bars = 95% CI).

interaction between *relationship type* and *empathic concern* did not greatly improve the model (QIC = 4140.908), and was non-significant ( $p = 0.132$). Adding in the three-way interaction between *relationship type*, *preference* and *empathic concern* did not improve the model (QIC = 4156.512), and was non-significant ( $p = 0.880$).

The model for political ideology is shown in table 5. There was a significant effect for 'rating type' such that compared with an overall rating of strong ($M = 4.06$ [95% CI = 3.94, 4.18]), fairness ($B = -0.26$, $p = 0.018$; 95% CI = −0.480, −0.044), compassion ($B = -0.84$, $p = 0.000$; 95% CI = −1.062, −0.621) and attractiveness ($B = -0.53$, $p = 0.000$; 95% CI = −0.771, −0.291), were all rated as significantly lower. The significant effect for *preferences* indicated that, compared with a preference to 'do nothing', overall ratings were more positive for preferences to 'both compensate and punish' ($B = 0.99$, $p = 0.006$; 95% CI = 0.283, 1.697) and 'compensate' ($B = 2.26$, $p = 0.000$; 95% CI = 1.597, 2.914). The interaction between *rating type* and *preference* is depicted in figure 3.

The interaction of *political ideology* with *preference* showed that the association between levels of political ideology and overall ratings of 'mate quality' were significantly negative for the 'compensation' preference ($B = -0.200$ $p = 0.003$; 95% CI = −0.334, −0.066) compared with the preference to 'do nothing'. Thus, greater left-wing political ideology is associated with an overall higher rating of 'mate quality' when a compensatory preference is used compared with 'do nothing'. All other comparisons for the interaction were non-significant. Adding in the additional interaction between *relationship type* and *empathic concern* did not greatly improve the model (QIC = 3878.687) and was non-significant ( $p = 0.275$). Adding in a three-way interaction between *relationship type*, *preference* and *empathic concern* did not improve the model (QIC = 3893.661), and was non-significant ( $p = 0.813$).

As a robustness check on the result reported in these regression models we dichotomized empathic concern and political ideology based on their median values and conducted a series of MANOVAs. The results replicate the findings reported in tables 4 and 5 (see electronic supplementary material, section D, figures S2 and S3). These results of Study Three did not vary as a function of relations status (single or not: electronic supplementary material, section D).

## 4. Discussion

In this paper we extend the work on the sexual selection of cooperation by female choice, by expanding the available cooperative choices to include punishment and compensation, both of which signal

**Table 3.** Factor loadings for ratings by preference for both short- and long-term relationship ($N = 159$). Columns refer to the loading of each judgement on its specified factor.

| | overall mate quality | | | |
| --- | --- | --- | --- | --- |
| | compensate and punish | do nothing | compensate | punish |
| *compensate and punish*: compassion | **0.88** ($p = 0.000$) | 0.00 | 00 | 00 |
| *compensate and punish*: attractive | **0.93** ($p = 0.000$) | 0.00 | 00 | 00 |
| *compensate and punish*: fair | **0.83** ($p = 0.000$) | 0.00 | 00 | 00 |
| *compensate and punish*: strong | **0.74** ($p = 0.000$) | 0.00 | 00 | 00 |
| *do nothing*: compassion | 00 | **0.89** ($p = 0.000$) | 00 | 00 |
| *do nothing*: attractive | 00 | **0.83** ($p = 0.000$) | 00 | 00 |
| *do nothing*: fair | 00 | **0.81** ($p = 0.000$) | 00 | 00 |
| *do nothing*: strong | 00 | **0.64** ($p = 0.000$) | 00 | 00 |
| *compensate*: compassion | 00 | 00 | **0.78** ($p = 0.000$) | 00 |
| *compensate*: attractive | 00 | 00 | **0.83** ($p = 0.000$) | 00 |
| *compensate*: fair | 00 | 00 | **0.75** ($p = 0.000$) | 00 |
| *compensate*: strong | 00 | 00 | **0.72** ($p = 0.000$) | 0.00 |
| *punish*: compassion | 00 | 00 | 00 | **0.86** ($p = 0.000$) |
| *punish*: attractive | 00 | 00 | 00 | **0.80** ($p = 0.000$) |
| *punish*: fair | 00 | 00 | 00 | **0.89** ($p = 0.000$) |
| *punish*: strong | 00 | 00 | 00 | **0.63** ($p = 0.000$) |
| $\alpha$ | 0.88 | 0.84 | 0.80 | 0.84 |
| | latent factor correlations | | | |
| compensate and punish | 1 | | | |
| do nothing | −0.030 ($p = 0.681$) | 1 | | |
| compensate | 0.070 ($p = 0.306$) | −0.418 ($p = 0.000$) | 1 | |
| punish | 0.746 ($p = 0.000$) | 0.009 ($p = 0.905$) | −0.071 ($p = 0.302$) | 1 |

different positive mate qualities (e.g. compassion and fairness respectively). In an undergraduate sample we explored if men would prefer to compensate or punish, when in a romantic mind-set, and which of these women would choose. The main findings show that having witnessed an unfair interaction men, in a romantic mind-set, prefer to adopt strategies that focus either purely on 'compensation' or a mixture of 'both compensation and punishment' rather than just punishment. Congruent with this, women prefer men who 'compensate' when considering both short-term and long-term relationships. Furthermore, higher levels of trait empathy were associated with men's preference to compensate and women's preference for men who compensate. Similarly a left-wing political ideology in women was associated with a preference for men who compensate. We explore the theoretical implications of these findings below.

## 4.1. Sexual selection and the evolution of cooperation

In this series of studies we examined men's behavioural preferences and women's choices for men with respect to cooperation. As such, these data speak only to models of sexual selection that are based on

**Table 4.** Generalized estimating equations for empathic concern, preference and rating type. Relationship type (0 = short-term, 1 = long-term); rating type (1 = attractive, 2 = compassionate, 3 = fair and 4 = strong); preference (1 = punish, 2 = compensate, 3 = both punish and compensate and 4 = do nothing). QIC = quasi likelihood under independence model criteria.

|  | Wald $\chi^2$ (d.f.) | $p$ |
| --- | --- | --- |
| relationship type | 0.85 (1) | 0.357 |
| rating type | 69.14 (3) | 0.000 |
| preference | 4.65 (3) | 0.199 |
| empathic concern | 3.54 (1) | 0.060 |
| relationship type * preference | 2.63 (3) | 0.452 |
| rating type * preference | 235.46 (9) | 0.000 |
| preference * empathic concern | 36.51 (3) | 0.000 |
| intercept | 381.45 (1) | 0.000 |
| $N$ (observations) | 160 (2548) | |
| QIC | 4146.853 | |

**Table 5.** Generalized estimating equations for political ideology, preference and rating type. Relationship type (0 = short-term, 1 = long-term); rating type (1 = attractive, 2 = compassionate, 3 = fair and 4 = strong); preference (1 = punish, 2 = compensate, 3 = both punish and compensate and 4 = do nothing). QIC = quasi likelihood under independence model criteria.

|  | Wald $\chi^2$ (d.f.) | $p$ |
| --- | --- | --- |
| relationship type | 0.82 (1) | 0.365 |
| rating type | 61.01 (3) | 0.000 |
| preference | 231.14 (3) | 0.000 |
| political ideology | 0.19 (1) | 0.662 |
| relationship type * preference | 1.98 (3) | 0.577 |
| rating type * preference | 213.29 (9) | 0.000 |
| preference * political ideology | 13.42 (3) | 0.004 |
| intercept | 1006.17 (1) | 0.000 |
| $N$ (observations) | 145 (2319) | |
| QIC | 3879.103 | |

female choice [10]. We acknowledge that sexual selection is a much more complex phenomenon than this [8,10] and suggest some future avenues for research below in §4.4.

When in a romantic mind-set, men who have the option to punish a transgressor or compensate a victim chose to either 'compensate' or 'both compensate and punish'. Congruently, when seeking a partner, women report finding men who chose a strategy that contains compensation as the most attractive, compassionate, fair and strong. This congruency is generally independent of the relationship type (short- versus long-term). With respect to the evolution of cooperation, sexually selecting on compensation, as a signal of self-serving mutually-beneficial behaviour, has a number of advantages. First, compensation is associated with greater perceptions of fairness by women. Fairness is a major component underlying reciprocity (both direct and indirect), and inequality aversion [67,68] both of which are key proximal mechanisms for cooperation [5]. Second, compensation is associated with greater perceptions of compassion [38] which is linked to empathy [64], again a key proximal component of cooperation. Furthermore, behaviours like compensation increase the probability of the recipient helping a stranger [56,106] and the actor being helped [107], both of which have the advantage of increasing the spread of cooperation via reciprocity [57]. Third, men's preference to compensate and a women's preference for men who compensate were both associated with higher levels of trait empathic concern, suggesting that empathic concern may be sexually selected due to its association with the display of and preferences for compensation [78]. The same is true of political ideology.

Interestingly, the preference for men to compensate is rated by women also as signalling strength. While it may be argued that strength is more likely signified by the use of punishment [56], these data show that when there is an option to punish and/or compensate, the choice to compensate is perceived by women as a signifier of strength. One possibility for this could be due to the direct other-regarding aspect of compensation: the victim is directly helped. This direct victim benefit is not present in self-serving punishment. Self-serving punishment does, however, have indirect other-regarding protective benefits in terms of potentially deterring future transgressions. It may be that it is the direct other-regarding component on compensation that signifies strength, especially when framed against the possibility of self-serving punishment. Choosing to help someone directly is seen as a 'stronger' thing to do than helping them indirectly via punishment.

There was some variation by relationship length (short-term versus long-term) in women's ratings. Women preferred *strength* for long-term relationships. This may reflect dominance and status that women often rate as desirable qualities in men for long-term relationship [11,69]. Women preferred attractive men for short-term relationships and again this may reflect physical attractiveness as a desirable quality when the relationship is seen as one that will not last [22,69]. Women found men who punish as the least unattractive for short-term compared with long-term relationships. Indeed, there is evidence that women prefer men with dark-triad traits (i.e. psychopathy, narcissism, Machiavellianism: so called 'cads' not 'dads') for short-term rather than long-term relationships [108]. Choosing to punish, when others compensate, may signify dark-triad traits in males. This may explain why women find punishers less unattractive for short-term relationships. However, this did not replicate in Study Three, so this effect should be treated with caution. Indeed, all these effects require replication.

## 4.2. Sexual selection or social selection

In this paper we have used sexual selection theory as our guiding model. However, recent debates in the literature have focused on *social selection* and whether sexual selection should be viewed as a sub-component of social selection [109] or indeed if social selection is an alternative to sexual selection [110]. Social selection occurs when the fitness of an individual is influenced by that individual's interaction with others, when competing for resources [109–112]. Sexual selection can be viewed as a type of social selection whereby the competition is for mates [109,112]. Thinking in this way means that body ornaments and display may arise from social behaviour that is not focused on sexual selection [109]. Social selection as an alternative to sexual selection, therefore, sees ornaments and displays as ways of negotiating social interactions and the formation of teams (which may be mating pairs) [110]. One possibility raised by this approach is that mutually-beneficial cooperation is pleasurable to the interacting parties [110]. Each interacting partner has both a self-serving and other-regarding preference and if they enjoy each other's welfare as much as their own, they will maximize their fitness by cooperating [110, pp. 2297–2298]. This scenario is very similar to scenarios described by Bshary & Bergmüller [24] where the actor and recipient can *influence each other's* outcomes which they term *mutual cooperation* (+/+) when this is with respect to *immediate* fitness outcomes and *cooperation (within species)* (+/+) when this is with respect to *lifetime* direct fitness outcomes. While we acknowledge that mutually-beneficial behaviour may have a joint pleasure function and this may be especially important when working on a joint task (including child rearing), our task, the 3PPC game, was not a joint task. Furthermore, our findings show that men alter their behaviour when in a romantic mind-set. This demonstration that the behaviour is different when in a romantic mind-set suggests that the *self-serving mutually-beneficial behaviour* studied here does have a sexual selection component [109], and as such indicates that in this context sexual selection is probably best thought of as a subset of social selection.

## 4.3. The role of punishment

While compensation is a strong signal of cooperative qualities in a mate it is not strong enough to elevate women's ratings of men who choose to punishment, to the level of pure compensation, when it is combined into a joint 'punish and compensate' strategy. Therefore, by using a mixture of compensation and punishment, men can enhance their attractiveness to women but not to the same extent as pure compensation. Again this may reflect the fact that self-serving punishment only delivers *indirect* other-regarding benefits. Spending resources to punish detracts from spending resources directly to help a victim. Thus, it may well be the case that women rate men who combine

compensation and punishment, as having a lower overall mate quality, as they are wasting resources punishing that could be used to directly help.

These findings do not mean that self-serving punishment is not an important force for the evolution and sustainability of cooperation [42,50]. Rather our results suggest that, when compared with compensation, it is not likely to be open to as strong a sexual selection effect based on female choice. Self-serving punishment may be more desirable in sexual selection context where males cooperate to complete for mates [17]. For example, if cooperation among males is important for securing the upbringing of offspring [17], then males who are able to increase levels of cooperation with punishment may be more desired.

## 4.4. Future directions

There are a number of interesting potential avenues, beyond a female choice model, for development of sexual selection and cooperation research. First, there is evidence that males may cooperate to win in the competition for sexual partners [17] and it would be interesting to explore how women perceive males who cooperate to achieve a goal (e.g. build up resources of food) versus those who choose to do this on their own. Exploring male choice and how women alter their displays of cooperation to attract males would be worthy of study [18,32]. The 3PPC game we used, is often used to explore ideas of restorative justice and it seems reasonable to extend the scope of the effects of sexual selection into the domain of moral decision making. Indeed, there is evidence that people who show a deontological moral preference are preferred as cooperative partners [113]. It would be interesting to know if this preference is open to sexual selection.

## 4.5. Limitations

While it has been argued that the study of sexual selection in human's can shed important light on how human traits evolve [114], it has to be acknowledged that all three studies reported in this paper are based on Western educated (undergraduate) samples [115], and that this may limit the generalizability of these findings [115]. Thus we feel we provide some initial evidence that sexual selection, based on female choice, *can* favour compensation over punishment when fairness norms have been violated. However, where cultural norms focus more on status or power, punishment may be the sexually selected preference over compensation. Indeed, there is evidence that levels of punishment for example vary cross-culturally [116]. Furthermore, the concept of cooperation is culturally specific and in many cultures it is community and kin based [117] rather than the focus on helping strangers as is common in the West. Thus, these findings require replication in larger more diverse samples.

With respect to the reaction of women we did not vary how much money men spent to punish, compensate, or do both, as in this series of studies we were concerned primarily with establishing the basic effects of the behavioural act rather than their relative cost. Indeed, there is good reason to believe that the act of helping or punishing is itself as strong a signal as the cost paid to help or punish [47]. Furthermore, women are more likely to be helped than men, and men more likely to help women than other men [118]. Therefore, the gender of the recipient and the actor needs to be considered. Thus, we feel we have established the basic behavioural response and further research can explore how this is affected by the cost men are willing to incur to punish, compensate, or do both, as well as their relative wealth and who the victim is.

It would also be useful to expand the ratings of mate quality to include intelligence, humour, wealth, and dominance, as well as physical and emotional attractiveness [11,12]. However, while it is not clear if the attractiveness ratings used here relate to physical attractiveness, there is some evidence that they do—more so for men than women [119]. However, it would be useful to differentiate physical attractiveness from other characteristics of attractiveness such as use of humour and social skills [119]. Similar issues arise concerning the assessment of strength that could represent either physical strength or 'strength of character'. Work in the area of 'strength of character' has identified the presence of an 'other-directed' character strength that contains kindness, modesty and bravery [120]. Consistent with this, the results from the factor analytic models reported in this paper show that ratings of strength loaded with ratings of fairness, compassion and attractiveness. Thus, we feel that strength in this context is about character virtues rather than physical strength. However, in future work it would be worth distinguishing these to see if the effect on an overall rating of strength observed here represents either physical strength, strength of

character or both. This is especially the case given the argument above that women attribute greater strength to men who compensate, due resources being given directly to the victim, rather than men who punish, as here resources go indirectly to helping the victim. Thus, resources spent on punishment may be seen as wasted when the option to compensate is present. This may represent strength of character more than physical strength.

Ethics. All studies were registered and approved according to the procedures of the ethics committee of the School of Psychology, University of Nottingham (registration numbers = 325, 404 and 503). All participants provided informed consent.

Data accessibility. The data files for the three studies can be accessed at the Dryad Digital Repository: https://doi.org/10.5061/dryad.738pm17 [121] The syntax for the analyses in the main text can be found in the electronic supplementary material, section E.

Authors' contributions. E.F. conceived of all three studies. E.F. and H.T. designed Studies One and Two. H.T. collected data for Study One and Study Two. F.H. collected data for Study Two. F.H. coded and analysed the free-responses for Study Two. E.F., G.P. and L.S. contributed the design of Study Three. E.Q., G.P. and L.S. collected data for Study Three. E.F. analysed the data and wrote the manuscript with comments from H.T., F.H., G.P., L.S. and E.Q. All authors approve the final version and agree to be accountable for its content.

Competing interests. We declare we have no competing interests.

Funding. The studies were funded by the School of Psychology, University of Nottingham.

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
