## [Reviewer comments · Royal Society Open Science]

Review History

Decision letter (RSOS-180216.R0)

28-Feb-2018

Dear Dr ferguson:

Manuscript ID RSOS-180216 entitled "Compassion Wins Out: Sexual Selection and Men and Women's Preferences for Punishment and Compensation" which you submitted to Royal Society Open Science, has been assessed. The comments from the editor are included at the bottom of this letter.

In view of the criticisms of the editor, the manuscript has been rejected in its current form. However, a new manuscript may be submitted which takes into consideration these comments.

Please note that resubmitting your manuscript does not guarantee eventual acceptance, and that your resubmission will be subject to peer review before a decision is made.

Your resubmitted manuscript should be submitted by 28-Aug-2018. If you are unable to submit by this date please contact the Editorial Office.

Please note that Royal Society Open Science will introduce article processing charges for all new submissions received from 1 January 2018. Charges will also apply to papers transferred to Royal Society Open Science from other Royal Society Publishing journals, as well as papers submitted as part of our collaboration with the Royal Society of Chemistry (<http://rsos.royalsocietypublishing.org/chemistry>). If your manuscript is submitted and accepted for publication after 1 Jan 2018, you will be asked to pay the article processing charge, unless you request a waiver and this is approved by Royal Society Publishing. You can find out more about the charges at <http://rsos.royalsocietypublishing.org/page/charges>. Should you have any queries, please contact openscience@royalsociety.org.

on behalf of Dr Molly Crockett (Associate Editor) and Antonia Hamilton (Subject Editor)
openscience@royalsociety.org

Associate Editor Comments to Author (Dr Molly Crockett):

Dear Drs Ferguson, Harrison and Tellentire,

Thank you for submitting your manuscript, "Compassion Wins Out: Sexual Selection and Men and Women's Preferences for Punishment and Compensation", to Royal Society Open Science. I read your manuscript and regret to inform you we will not be able to send it out for peer review. Below I outline some issues with the manuscript that formed the basis of this decision.

First, I expected to see a more comprehensive literature review and depth of scholarship. The introduction is just over 300 words, with 12 references. As there is an extensive literature on sexual selection and punishment, much more needs to be done to motivate the current study in the context of prior relevant literature. A similar criticism applies to the discussion, at less than 400 words and 8 references.

Second, there are issues with the methodology. The sample size for Experiment 1 is not well

justified. The experimental conditions in Experiment 2 are not counterbalanced, which strongly limits the conclusions that can be drawn. I also have concerns that a (non-counterbalanced) within-subjects 2 x 4 design may alert participants to the hypotheses of the study, and it is not clear whether anything was done to counter possible demand effects. I would also recommend double-checking the reported statistics as there seem to be some implausible values (e.g. on pg 7 line 142).

Finally, given how short the paper is, it was unclear why many of the analyses were placed in supplementary materials, including analyses designed to test central hypotheses around the role of compassion.

I would be willing to reconsider a new submission that addresses the above concerns, however it would need to include a much more comprehensive literature review and discussion that describes in more detail previous work on this topic, providing context for the new studies, a more comprehensive results section, and a replication of the key findings with a fully counterbalanced and ideally between-subjects design, before I can justify sending this out for further peer review.

I wish you the best of luck with your research. Thank you for considering our journal as an outlet for your work.

Author's Response to Decision Letter for (RSOS-180216.R0)

See Appendix A.

RSOS-181441.R0

Review form: Reviewer 1

Is the manuscript scientifically sound in its present form?

Yes

Are the interpretations and conclusions justified by the results?

Yes

Is the language acceptable?

Yes

Is it clear how to access all supporting data?

Yes

Do you have any ethical concerns with this paper?

No

Have you any concerns about statistical analyses in this paper?

No

Recommendation?

Accept with minor revision (please list in comments)

Comments to the Author(s)

This paper explores the sexual selection of human pro-sociality using a version of a dictator game, in which a third party can compensate the victim of inequitable allocations, punish the dictator, do both, or do nothing. Study 1 shows that romantically-primed men more often chose to compensate or chose a combination of punishment and compensation compared to neutrally primed men. Study 2 shows that women rate men who chose compensation or compensation and punishment, in the decision situation above, more attractive compared to men who chose punishment or doing nothing. Participants were first asked to rate attractiveness when considering the person for a short-term relationship and afterwards when considering the person for a long-term relationship. Study 3 replicates study 2, randomly allocating participants to either the short-term or the long-term relationship conditions, as well as including ratings of indicators of cooperativeness and social status. It also investigates trait empathy and political ideology as correlates with preferences for compensation or punishment.

Comments and suggestions:

Page 7, line 136: The authors write 'changing the choice architecture alters how respondents make choices' citing Thaler and Sunstein (2009)'s 'Nudge'. A further clarification is necessary how this relates to the introduction of third-party punishment to a game in which a third party can help/not help. Including punishment seems to be much more than a gentle 'nudge', changing the game's strategies, possible payoffs and, depending on expected use of punishment, potentially the payoff-maximising strategy.

Page 12, line 252: Did you mean the 'conditional information lottery' as described in Bardsley (2003), which is citation no. 68? I don't fully understand how this procedure was implemented here. A clarification would be very helpful. Please also describe what participants (in the role of player C) were told about players A and B. My main concern is: did the participants believe that their choices would have real payoff consequences for the other players?

Page 12, line 256: '8 participants would be randomly selected and paid based on their decision'. However, the instructions on page 2 of the Supplementary Files state: 'At the end of the whole experiment 5 participants will be randomly selected and they will be paid based on their decision in the task below'. How many participants were actually paid?

Page 14, line 299: 'Women were presented with the same 3PCC game (individually or in a group setting: 44% vs 56%)'. Did you implement different conditions (i.e., individual choice vs a group choice)? An explanation would be very helpful.

Page 14, line 303: Participants rated the attractiveness 'if he decided to spend his money on (1) punish player A, (2) compensate player B, (3) both punish and compensate or (4) keep the money'. What information was given to participants regarding the extent of compensation/punishment chosen by player C? For example, if player C would sacrifice the whole endowment to completely destroy player A's payoff, this might be viewed as a very extreme action (and a potential signal of dark-triad traits), especially in case a less extreme punishment was available.

Please proofread and correct typos throughout the paper.

Review form: Reviewer 2

Is the manuscript scientifically sound in its present form?

Yes

Are the interpretations and conclusions justified by the results?

No

Is the language acceptable?

Yes

Is it clear how to access all supporting data?

No

Do you have any ethical concerns with this paper?

No

Have you any concerns about statistical analyses in this paper?

I do not feel qualified to assess the statistics

Recommendation?

Major revision is needed (please make suggestions in comments)

Comments to the Author(s)

This is an interesting set of studies. It's an interesting question: whether sexual selection has played a role in the evolution of altruism and punishment. I particularly like that the authors have tested both sides of the signaling – the effect of mating motives on compensation & punishment, and the effect of compensation & punishment on mate preferences. Both sides tell a similar story: mating motives have an effect on men's willingness to help others but not punish wrongdoers, and women prefer men who help others but not who punish wrongdoers. My main question is about the theoretical contribution and conceptual difference between altruism and compensation.

MAIN POINT:

Many authors have shown that altruism might have been selected for: men are more generous in mating situations, and women prefer generous men to neutral men. The authors show something similar, but by contrast, the authors stress that their results are about **compensation** and redistributive justice, rather than altruism per se. This is their main claim for novelty. Altruism and compensation are two very similar concepts: one individual pays a cost to confer benefits on another. It's not obvious why it's important to discriminate between them. Altruism occurs most when one individual has some need (e.g., empty belly), such that the help being conferred (e.g., one unit of food) increases the recipient's fitness more than it decreases the actor's fitness (i.e., $b > c$). The major difference between altruism and compensation is that in compensation, the need occurs because another individual has been unfair. By contrast, altruism is more general in that the need can come from any source (e.g., bad harvests), rather than just from exploitative conspecifics. Thus, compensation is a particular subset of altruism, where an actor confers benefits upon a recipient to alleviate a need caused by someone else's unfairness.

If the authors wish to show that their results reflect compensation instead of altruism more generally, they will need to do both a) show that compensation is not a subset of altruism or generosity, and b) differentiate between them methodologically. In the current study, when participants compensate victims of unfairness, this is indistinguishable from generosity – one

individual confers benefits upon another. The current methodology does not distinguish compensation from altruism (or generosity or whatever else one wishes to call it), and there is no way to retroactively do so. Thus, the results are just as consistent with "sexual selection for altruism" as they are with "sexual selection for compensation or redistributive justice". The former is relatively well-established, so it would not be new, but that is not a problem if RSOS publishes any study that is methodologically sound regardless of novelty (like PLoS ONE does). If the authors wish to claim that their study specifically looks at compensation (and NOT altruism), then their methods will have to do something to differentiate between these two motivations. Otherwise, they will have to rewrite the sections that stress the sexual selection of compensation, and be upfront that their results are equally consistent with (the well-established) sexual selection of altruism. So in my opinion, the authors have two choices for publication: 1) reframe the manuscript to reduce the stress on redistributive justice and write it in terms of altruism; or 2) run a new study that differentiates between altruism and compensation. If novelty is not a deciding factor in publication at RSOS, then I would recommend the former (unless the authors really are most interesting in speaking about compensation). This reframing will need to be done in the introduction and the discussion (and abstract).

If by contrast the main contribution of this manuscript is about "resolving the tension between punishing and compensating", then I'm less convinced that that's an important question. I see the main contribution is in showing that punishment is not preferred in mating, but altruism is. That would be a useful addition to the literature.

SECONDARY POINT

On page 6 (lines 111-127), the authors attempt to argue that restorative justice is important because it relates to parenting skill. This is a stretch. I found this section unconvincing. Why is restorative justice particularly relevant here, beyond other forms of altruism? Why should punishment of unrelated third parties be related to demanding fairness among one's own children? (Indeed, mothers might be the people who do the most enforcement of fairness among their own children, but do the least third-party punishing.) To be honest, this sounded like post-hoc hypothesizing on why restorative justice is particularly relevant above and beyond other forms of altruism. As such, this section requires either significant strengthening or elimination.

TIERTIARY POINT

The analyses in section 3.3.5 are very hard to understand. This needs to be clarified. Similarly, Table 2 is hard to understand. What are the columns? How that trait loads on preference (for male type)... as measured by what? It seems a bit circular, that something about compensate and punishment would load heavily on compensate and punishment. If these are factors, then don't label them as male types. At the very least, this Table needs to be much better explained.

Tables 3 and 4 could be clarified a bit as well.

MINOR POINTS

It's a bit ambitious to say that the authors are aiming for a power of 80% to detect Cohen's $d=0.8$ (large effect size). If the authors wish to say that they had sufficient power to detect an effect of punishment, then they should have done the power analysis with a more realistic effect size (small-medium of 0.35 or medium of 0.5). At this point, now that the data are collected, the authors should do some sort of sensitivity analysis or post hoc power analysis: how big would the effect size have to be to have an 80% chance of detecting it with their sample size? Or something similar, to show that the lack of significance is not just due to low power.

In Study 1, how much time was there between the prime and the dependent measure? Is it likely that the prime would last that long?

Is there an overall negative effect of punishment in Study 3?

There are multiple typos, including in the Figure 1 caption.

Decision letter (RSOS-181441.R0)

12-Feb-2019

Dear Dr Ferguson,

The Subject Editor assigned to your paper ("Compassion Wins Out: Sexual Selection and Men and Women's Preferences for Punishment and Compensation") has now received comments from reviewers. We would like you to revise your paper in accordance with the referee and Associate Editor suggestions which can be found below (not including confidential reports to the Editor). Please note this decision does not guarantee eventual acceptance.

Please submit a copy of your revised paper before 07-Mar-2019. Please note that the revision deadline will expire at 00.00am on this date. If we do not hear from you within this time then it will be assumed that the paper has been withdrawn. In exceptional circumstances, extensions may be possible if agreed with the Editorial Office in advance. We do not allow multiple rounds of revision so we urge you to make every effort to fully address all of the comments at this stage. If deemed necessary by the Editors, your manuscript will be sent back to one or more of the original reviewers for assessment. If the original reviewers are not available we may invite new reviewers.

When submitting your revised manuscript, you must respond to the comments made by the referees and upload a file "Response to Referees" in "Section 6 - File Upload". Please use this to document how you have responded to each of the comments, and the adjustments you have made. In order to expedite the processing of the revised manuscript, please be as specific as possible in your response.

- Ethics statement

- Data accessibility

It is a condition of publication that all supporting data are made available either as supplementary information or preferably in a suitable permanent repository. The data accessibility section should state where the article's supporting data can be accessed. This section

should also include details, where possible of where to access other relevant research materials such as statistical tools, protocols, software etc can be accessed. If the data has been deposited in an external repository this section should list the database, accession number and link to the DOI for all data from the article that has been made publicly available. Data sets that have been deposited in an external repository and have a DOI should also be appropriately cited in the manuscript and included in the reference list.

If you wish to submit your supporting data or code to Dryad (<http://datadryad.org/>), or modify your current submission to dryad, please use the following link:
<http://datadryad.org/submit?journalID=RSOS&manu=RSOS-181441>

- **Competing interests**

- **Authors' contributions**

- **Acknowledgements**

- **Funding statement**

on behalf of Prof Antonia Hamilton (Subject Editor)
openscience@royalsociety.org

Reviewer comments to Author:

Reviewer: 1

Comments to the Author(s)

This paper explores the sexual selection of human pro-sociality using a version of a dictator game, in which a third party can compensate the victim of inequitable allocations, punish the dictator, do both, or do nothing. Study 1 shows that romantically-primed men more often chose to compensate or chose a combination of punishment and compensation compared to neutrally primed men. Study 2 shows that women rate men who chose compensation or compensation and punishment, in the decision situation above, more attractive compared to men who chose punishment or doing nothing. Participants were first asked to rate attractiveness when considering the person for a short-term relationship and afterwards when considering the person for a long-term relationship. Study 3 replicates study 2, randomly allocating participants to either the short-term or the long-term relationship conditions, as well as including ratings of indicators of cooperativeness and social status. It also investigates trait empathy and political ideology as correlates with preferences for compensation or punishment.

Comments and suggestions:

Page 7, line 136: The authors write 'changing the choice architecture alters how respondents make choices' citing Thaler and Sunstein (2009)'s 'Nudge'. A further clarification is necessary how this relates to the introduction of third-party punishment to a game in which a third party can help/not help. Including punishment seems to be much more than a gentle 'nudge', changing the game's strategies, possible payoffs and, depending on expected use of punishment, potentially the payoff-maximising strategy.

Page 12, line 252: Did you mean the 'conditional information lottery' as described in Bardsley (2003), which is citation no. 68? I don't fully understand how this procedure was implemented here. A clarification would be very helpful. Please also describe what participants (in the role of player C) were told about players A and B. My main concern is: did the participants believe that their choices would have real payoff consequences for the other players?

Page 12, line 256: '8 participants would be randomly selected and paid based on their decision'. However, the instructions on page 2 of the Supplementary Files state: 'At the end of the whole experiment 5 participants will be randomly selected and they will be paid based on their decision in the task below'. How many participants were actually paid?

Page 14, line 299: 'Women were presented with the same 3PCC game (individually or in a group setting: 44% vs 56%)'. Did you implement different conditions (i.e., individual choice vs a group choice)? An explanation would be very helpful.

Page 14, line 303: Participants rated the attractiveness 'if he decided to spend his money on (1) punish player A, (2) compensate player B, (3) both punish and compensate or (4) keep the money'. What information was given to participants regarding the extent of compensation/punishment chosen by player C? For example, if player C would sacrifice the whole endowment to completely destroy player A's payoff, this might be viewed as a very extreme action (and a potential signal of dark-triad traits), especially in case a less extreme punishment was available.

Please proofread and correct typos throughout the paper.

Reviewer: 2

Comments to the Author(s)

This is an interesting set of studies. It's an interesting question: whether sexual selection has played a role in the evolution of altruism and punishment. I particularly like that the authors have tested both sides of the signaling – the effect of mating motives on compensation & punishment, and the effect of compensation & punishment on mate preferences. Both sides tell a similar story: mating motives have an effect on men's willingness to help others but not punish wrongdoers, and women prefer men who help others but not who punish wrongdoers. My main question is about the theoretical contribution and conceptual difference between altruism and compensation.

MAIN POINT:

Many authors have shown that altruism might have been selected for: men are more generous in mating situations, and women prefer generous men to neutral men. The authors show something similar, but by contrast, the authors stress that their results are about *compensation* and redistributive justice, rather than altruism per se. This is their main claim for novelty. Altruism and compensation are two very similar concepts: one individual pays a cost to confer benefits on another. It's not obvious why it's important to discriminate between them. Altruism occurs most when one individual has some need (e.g., empty belly), such that the help being conferred (e.g., one unit of food) increases the recipient's fitness more than it decreases the actor's fitness (i.e., $b > c$). The major difference between altruism and compensation is that in compensation, the need occurs because another individual has been unfair. By contrast, altruism is more general in that the need can come from any source (e.g., bad harvests), rather than just from exploitative conspecifics. Thus, compensation is a particular subset of altruism, where an actor confers benefits upon a recipient to alleviate a need caused by someone else's unfairness.

If the authors wish to show that their results reflect compensation instead of altruism more generally, they will need to do both a) show that compensation is not a subset of altruism or generosity, and b) differentiate between them methodologically. In the current study, when participants compensate victims of unfairness, this is indistinguishable from generosity – one individual confers benefits upon another. The current methodology does not distinguish compensation from altruism (or generosity or whatever else one wishes to call it), and there is no way to retroactively do so. Thus, the results are just as consistent with "sexual selection for altruism" as they are with "sexual selection for compensation or redistributive justice". The former is relatively well-established, so it would not be new, but that is not a problem if RSOS publishes any study that is methodologically sound regardless of novelty (like PLoS ONE does). If the authors wish to claim that their study specifically looks at compensation (and NOT altruism), then their methods will have to do something to differentiate between these two motivations. Otherwise, they will have to rewrite the sections that stress the sexual selection of compensation, and be upfront that their results are equally consistent with (the well-established) sexual selection of altruism. So in my opinion, the authors have two choices for publication: 1) reframe the manuscript to reduce the stress on redistributive justice and write it in terms of altruism; or 2) run a new study that differentiates between altruism and compensation. If novelty is not a deciding factor in publication at RSOS, then I would recommend the former (unless the authors really are most interesting in speaking about compensation). This reframing will need to be done in the introduction and the discussion (and abstract).

If by contrast the main contribution of this manuscript is about "resolving the tension between punishing and compensating", then I'm less convinced that that's an important question. I see the

main contribution is in showing that punishment is not preferred in mating, but altruism is. That would be a useful addition to the literature.

SECONDARY POINT

On page 6 (lines 111-127), the authors attempt to argue that restorative justice is important because it relates to parenting skill. This is a stretch. I found this section unconvincing. Why is restorative justice particularly relevant here, beyond other forms of altruism? Why should punishment of unrelated third parties be related to demanding fairness among one's own children? (Indeed, mothers might be the people who do the most enforcement of fairness among their own children, but do the least third-party punishing.) To be honest, this sounded like post-hoc hypothesizing on why restorative justice is particularly relevant above and beyond other forms of altruism. As such, this section requires either significant strengthening or elimination.

TIERTIARY POINT

The analyses in section 3.3.5 are very hard to understand. This needs to be clarified. Similarly, Table 2 is hard to understand. What are the columns? How that trait loads on preference (for male type)... as measured by what? It seems a bit circular, that something about compensate and punishment would load heavily on compensate and punishment. If these are factors, then don't label them as male types. At the very least, this Table needs to be much better explained.

Tables 3 and 4 could be clarified a bit as well.

MINOR POINTS

It's a bit ambitious to say that the authors are aiming for a power of 80% to detect Cohen's $d=0.8$ (large effect size). If the authors wish to say that they had sufficient power to detect an effect of punishment, then they should have done the power analysis with a more realistic effect size (small-medium of 0.35 or medium of 0.5). At this point, now that the data are collected, the authors should do some sort of sensitivity analysis or post hoc power analysis: how big would the effect size have to be to have an 80% chance of detecting it with their sample size? Or something similar, to show that the lack of significance is not just due to low power.

In Study 1, how much time was there between the prime and the dependent measure? Is it likely that the prime would last that long?

Is there an overall negative effect of punishment in Study 3?

There are multiple typos, including in the Figure 1 caption.

Author's Response to Decision Letter for (RSOS-181441.R0)

See Appendix B.

RSOS-181441.R1 (Revision)

Review form: Reviewer 3

Is the manuscript scientifically sound in its present form?

No

Are the interpretations and conclusions justified by the results?

No

Is the language acceptable?

No

Is it clear how to access all supporting data?

No

Do you have any ethical concerns with this paper?

No

Have you any concerns about statistical analyses in this paper?

I do not feel qualified to assess the statistics

Recommendation?

Major revision is needed (please make suggestions in comments)

Comments to the Author(s)

I think that the authors have interesting data here, and I like that they looked at both the signaling side (here, male behavior in a simulated mating context) and the signal receiver side (here, female preference for men as romantic partners, contingent on male behavior). I want to see these data published, but I find it very difficult to interpret this work as currently written so I recommend revision to enhance clarity.

Two problems, perhaps related, jump out at me immediately: inconsistent terminology, and unclear connections to functional theories about mating psychology and cooperation more generally. I further describe these two major problems, then note some other issues.

Major problem 1: Definitions of key terms

The central concepts of the paper are not clearly defined, and key terms seem to be used in different ways in different sections. Altruism is introduced by way of example rather than definition. Altruism at first seems to describe behavior -- helping, generosity, etc. Then in line 65 altruism seems to be an individual trait that is signaled by helping behavior. The usage goes back and forth after this.

Furthermore, punishment is introduced as "altruistic" and called a "key mechanism for the evolution and survival of altruism," but then punishment is contrasted with "altruism" for the rest of the paper. This is extremely confusing. For coordinating terminology, especially for those of us trying to link human psychology to biological theory, I always recommend West et al 2007 (<https://doi.org/10.1111%2Fj.1420-9101.2006.01258.x>). Other semantic schemes could also work, as long as terms are used in a clear, consistent way.

Major problem 2: Tenuous links to evolutionary theories

In the opening paragraph, the authors assert that sexual selection on altruistic behavior must involve female preference for male behavior; this ignores the obvious logical possibility of selection by male preference for female behavior. It is fine for the authors to only investigate one direction of sexual selection; it is not fine to claim the other direction does not exist. Doing so in the opening paragraph sets an unfortunate tone that the connection to relevant work might be too superficial. Indeed, the authors cite a very limited subset of relevant literature in support of their errant claim. The authors go on to cite other relevant literature, but haven't synthesized its relevance to the current point. I urge the authors to engage more deeply with recent theory and evidence. And, again, the confusion about terminology described above, makes it hard to connect the present work to ultimate theories about the evolution of (certain types of) cooperation, versus proximate theories about mechanisms involved in cooperative behavior (e.g., empathy), etc.

Other notes:

I see that data and materials are public. I do not see analysis code shared anywhere; apologies if I'm overlooking this.

Generally, I don't understand why many of these measures and methods are being used. The introduction doesn't adequately prepare me when the methods are presented.

"Attractiveness" seems intended to mean physical appearance, but unclear whether participants might interpret it more broadly. Same with "strength," which seems intended to mean physical muscular strength, but could be interpreted to include "strength of character" or the like.

Study 1 -- So participants are basically paying real money to send a signal to experimenter? Do we need to be concerned about personal characteristics of the researcher(s) and how the participants responded to the real human interaction? Did the same experimenter administer all sessions?

Study 2 -- Why is short/long the right dichotomy here? It seems more like serious/casual might be more appropriate. Is public/private an experimental variable? Is this an "experiment"?

Results 3.3.4 and 3.3.5 are very hard to follow.

Decision letter (RSOS-181441.R1)

17-Apr-2019

Dear Dr Ferguson:

Manuscript ID RSOS-181441.R1 entitled "To Help or Punish in the Face of Unfairness: Men and Women Prefer Altruism over Punishment in a Sexual Selection Context" which you submitted to Royal Society Open Science, has been reviewed. The comments of the reviewer(s) are included at the bottom of this letter.

Please submit a copy of your revised paper before 10-May-2019. Please note that the revision deadline will expire at 00.00am on this date. If we do not hear from you within this time then it will be assumed that the paper has been withdrawn. In exceptional circumstances, extensions may be possible if agreed with the Editorial Office in advance. We do not allow multiple rounds

of revision so we urge you to make every effort to fully address all of the comments at this stage. If deemed necessary by the Editors, your manuscript will be sent back to one or more of the original reviewers for assessment. If the original reviewers are not available we may invite new reviewers.

- Ethics statement

- Data accessibility

- Competing interests

- Authors' contributions

- Acknowledgements

- Funding statement

on behalf of Professor Antonia Hamilton (Subject Editor)
openscience@royalsociety.org

Editor's Comments to Author:

As the original reviewers of your RSOS-181441 were unavailable, a new referee has commented on your work. They see merit in the publication; however, they also point out a number of problems with the way it is presented (among other matters). As the data is interesting, we will allow one further revision but must warn that if the reviewers and editors are not happy with the clarity of the writing and links to theories (see comments below), the paper will be rejected with no further option to resubmit.

Reviewer comments to Author:

Reviewer: 3

Comments to the Author(s)

I think that the authors have interesting data here, and I like that they looked at both the signaling side (here, male behavior in a simulated mating context) and the signal receiver side (here, female preference for men as romantic partners, contingent on male behavior). I want to see these data published, but I find it very difficult to interpret this work as currently written so I recommend revision to enhance clarity.

Two problems, perhaps related, jump out at me immediately: inconsistent terminology, and unclear connections to functional theories about mating psychology and cooperation more generally. I further describe these two major problems, then note some other issues.

Major problem 1: Definitions of key terms

The central concepts of the paper are not clearly defined, and key terms seem to be used in different ways in different sections. Altruism is introduced by way of example rather than definition. Altruism at first seems to describe behavior -- helping, generosity, etc. Then in line 65

altruism seems to be an individual trait that is signaled by helping behavior. The usage goes back and forth after this.

Furthermore, punishment is introduced as "altruistic" and called a "key mechanism for the evolution and survival of altruism," but then punishment is contrasted with "altruism" for the rest of the paper. This is extremely confusing. For coordinating terminology, especially for those of us trying to link human psychology to biological theory, I always recommend West et al 2007 (<https://doi.org/10.1111%2Fj.1420-9101.2006.01258.x>). Other semantic schemes could also work, as long as terms are used in a clear, consistent way.

Major problem 2: Tenuous links to evolutionary theories

In the opening paragraph, the authors assert that sexual selection on altruistic behavior must involve female preference for male behavior; this ignores the obvious logical possibility of selection by male preference for female behavior. It is fine for the authors to only investigate one direction of sexual selection; it is not fine to claim the other direction does not exist. Doing so in the opening paragraph sets an unfortunate tone that the connection to relevant work might be too superficial. Indeed, the authors cite a very limited subset of relevant literature in support of their errant claim. The authors go on to cite other relevant literature, but haven't synthesized its relevance to the current point. I urge the authors to engage more deeply with recent theory and evidence. And, again, the confusion about terminology described above, makes it hard to connect the present work to ultimate theories about the evolution of (certain types of) cooperation, versus proximate theories about mechanisms involved in cooperative behavior (e.g., empathy), etc.

Other notes:

I see that data and materials are public. I do not see analysis code shared anywhere; apologies if I'm overlooking this.

Generally, I don't understand why many of these measures and methods are being used. The introduction doesn't adequately prepare me when the methods are presented.

"Attractiveness" seems intended to mean physical appearance, but unclear whether participants might interpret it more broadly. Same with "strength," which seems intended to mean physical muscular strength, but could be interpreted to include "strength of character" or the like.

Study 1 -- So participants are basically paying real money to send a signal to experimenter? Do we need to be concerned about personal characteristics of the researcher(s) and how the participants responded to the real human interaction? Did the same experimenter administer all sessions?

Study 2 -- Why is short/long the right dichotomy here? It seems more like serious/casual might be more appropriate. Is public/private an experimental variable? Is this an "experiment"?

Results 3.3.4 and 3.3.5 are very hard to follow.

Author's Response to Decision Letter for (RSOS-181441.R1)

See Appendix C.

RSOS-181441.R2 (Revision)

Review form: Reviewer 3

Is the manuscript scientifically sound in its present form?

Yes

Are the interpretations and conclusions justified by the results?

No

Is the language acceptable?

No

Do you have any ethical concerns with this paper?

No

Have you any concerns about statistical analyses in this paper?

No

Recommendation?

Accept with minor revision (please list in comments)

Comments to the Author(s)

As I said before, I like a lot of what the authors did here. The 4-choice design is an excellent extension of prior work, and looking at both the signaler and receiver side is great. The authors have made much clearer connections to the relevant literature in this version, which is much appreciated.

And, as before, I want to see these data published, but the manuscript in its present form is not yet suitable for publication. I list some concerns below. I believe that they can be addressed with minor revision.

1.

First, this work is based on ~350 British undergraduate participants. I appreciate that Line 747-749 acknowledges that mating-related preferences might vary with other aspects of social systems, and that Section 4.4 discusses cross-cultural considerations. But I am concerned that these crucial caveats are undermined with inflated claims, e.g. "We provide some initial evidence that sexual selection, based on female choice, favours compensation over punishment" or when the abstract simply says "We test which is preferred by men and chosen by women," etc. I think the authors have shown what sexual selection *can* favour, and they've shown what is preferred and chosen *by* undergrads at their University.* There are other places where the language is similarly excessive in my judgment.

2.

Next, and less substantively, proofreading is badly needed. I noticed lots of errors of spelling, grammar, punctuation, etc. I'm sympathetic, being prone to such errors myself. Here are some:

- "with respect [to]" (lines 80, 126)
- "High levels [of] cooperation" (line 125)
- "self-severing" lines 138, 180
- "Empathy is ... a mechanism[s]" lines 242-3
- "in two context[s]" line 379

- “overall ratings of the targets attractiveness” line 557

3.

I previously noted that there was no real person being punished/compensated, and suggested that the only way paying to alter the payoffs of a fictional character wasn't a total waste is if participants were trying to signal to the experimenter, in which case maybe we need to think about the participant-experimenter dynamic. I'm fine with the authors' response regarding anonymity, but I do need to raise another concern with a fake Player A: From the perspective of the female rater, why shouldn't we expect her to think that men who just give away their money to punish or compensate non-existent characters aren't the best partners? Shouldn't “do nothing” be the most attractive option, at least for a subset of women who see through the fiction? (Would presumably apply in the broader social selection framework too – do we want an ally who can't distinguish reality from fiction?) I don't think that another study with real stakes is required, but this concern should be discussed. (And it isn't totally clear that the women knew that the men were punishing/compensating a fake Player A. If they didn't know, this concern disappears, although then we might need to discuss the use of deception, unless the hypothetical-ness was clear.)

4.

“Women preferred strength for long-term relationships... Women preferred attractive men for short-term relationships” – I don't understand what these claims are based on. Women rated hypothetical men's strength and attractiveness as a function of the men's decision and the women's relationship context. So we can know which male actions women prefer, but how do we know which traits they prefer in different contexts? Aren't the traits entirely based on the actions?

5.

In general, since 3.3.4 suggests that all of the ratings load to some “mate quality” factor, and since “strength” seems to have a fuzzy meaning as deployed here, it is worth trying to parse the meanings of each individual rating word?

But if we want to do that, it seems to me that punishment could signal literal physical strength when the mode of punishment involves physical action (e.g., detaining and/or assaulting the perpetrator), whereas the mode of (fake) punishment here was economic. The first part in the general discussion about “strength” (line 682-92) was entirely based on a metaphorical notion of the word – ignoring the primary (somatic) definition of the word seems odd.

6.

Table 1 and related text: Could punishment have long-term reputation costs, e.g. a reputation for violence or cruelty? (And, as mentioned previously, could punishment or compensation of fictional characters have reputations costs?)

7.

From footnote 1, it sounds like the Study 1 “active control” condition participants were run months after the others. I'd suggest stating that explicitly rather than leaving it implicit.

8.

I don't object to the content of section 4.2, but it seems unnecessary in this paper, given that the methods were clearly aimed at mating psychology, rather than a more general cooperative psychology. Acknowledging that sexual selection can be seen as a subset of social selection (or perhaps other interpretations) and crediting important work on this topic could be accomplished far more efficiently, probably in the Intro.

9.

I suspect that the paper would read more smoothly if a lot of the technical details of 3.3.4 and maybe 3.3.5 were relegated to Supplement, with only summaries of that stuff in the main text.

10.

Figure text is difficult to read. Could be the journal system's auto-PDF distorting it?

Decision letter (RSOS-181441.R2)

14-Jun-2019

Dear Dr Ferguson:

On behalf of the Editors, I am pleased to inform you that your Manuscript RSOS-181441.R2 entitled "To Help or Punish in the Face of Unfairness: Men and Women Prefer Mutually-Beneficial Strategies over Punishment in a Sexual Selection Context" has been accepted for publication in Royal Society Open Science subject to minor revision in accordance with the referee suggestions. Please find the referees' comments at the end of this email.

The reviewers and Subject Editor have recommended publication, but also suggest some minor revisions to your manuscript. Therefore, I invite you to respond to the comments and revise your manuscript.

- Ethics statement

- Data accessibility

If you wish to submit your supporting data or code to Dryad (<http://datadryad.org/>), or modify your current submission to dryad, please use the following link:
<http://datadryad.org/submit?journalID=RSOS&manu=RSOS-181441.R2>

- Competing interests

- Authors' contributions

All submissions, other than those with a single author, must include an Authors' Contributions section which individually lists the specific contribution of each author. The list of Authors

should meet all of the following criteria; 1) substantial contributions to conception and design, or acquisition of data, or analysis and interpretation of data; 2) drafting the article or revising it critically for important intellectual content; and 3) final approval of the version to be published.

- Acknowledgements

- Funding statement

Because the schedule for publication is very tight, it is a condition of publication that you submit the revised version of your manuscript before 23-Jun-2019. Please note that the revision deadline will expire at 00.00am on this date. If you do not think you will be able to meet this date please let me know immediately.

- 1) A text file of the manuscript (tex, txt, rtf, docx or doc), references, tables (including captions) and figure captions. Do not upload a PDF as your "Main Document".
- 2) A separate electronic file of each figure (EPS or print-quality PDF preferred (either format should be produced directly from original creation package), or original software format)
- 3) Included a 100 word media summary of your paper when requested at submission. Please ensure you have entered correct contact details (email, institution and telephone) in your user account

4) Included the raw data to support the claims made in your paper. You can either include your data as electronic supplementary material or upload to a repository and include the relevant doi within your manuscript

5) All supplementary materials accompanying an accepted article will be treated as in their final form. Note that the Royal Society will neither edit nor typeset supplementary material and it will be hosted as provided. Please ensure that the supplementary material includes the paper details where possible (authors, article title, journal name).

on behalf of Antonia Hamilton (Subject Editor)
openscience@royalsociety.org

Editor Comments to Author:

The reviewer of this version of your manuscript has made a number of suggestions and comments - we would like to see these incorporated into a final version of your paper, and for you to provide a point-by-point response detailing how you've tackled these suggestions.

Reviewer comments to Author:
Reviewer: 3

Comments to the Author(s)

As I said before, I like a lot of what the authors did here. The 4-choice design is an excellent extension of prior work, and looking at both the signaler and receiver side is great. The authors have made much clearer connections to the relevant literature in this version, which is much appreciated.

And, as before, I want to see these data published, but the manuscript in its present form is not yet suitable for publication. I list some concerns below. I believe that they can be addressed with minor revision.

1.

First, this work is based on ~350 British undergraduate participants. I appreciate that Line 747-749 acknowledges that mating-related preferences might vary with other aspects of social systems, and that Section 4.4 discusses cross-cultural considerations. But I am concerned that

these crucial caveats are undermined with inflated claims, e.g. “We provide some initial evidence that sexual selection, based on female choice, favours compensation over punishment” or when the abstract simply says “We test which is preferred by men and chosen by women,” etc. I think the authors have shown what sexual selection *can* favour, and they’ve shown what is preferred and chosen *by* undergrads at their University.* There are other places where the language is similarly excessive in my judgment.

2.

Next, and less substantively, proofreading is badly needed. I noticed lots of errors of spelling, grammar, punctuation, etc. I’m sympathetic, being prone to such errors myself. Here are some:

- “with respect [to]” (lines 80, 126)
- “High levels [of] cooperation” (line 125)
- “self-severing” lines 138, 180
- “Empathy is ... a mechanism[s]” lines 242-3
- “in two context[s]” line 379
- “overall ratings of the targets attractiveness” line 557

3.

I previously noted that there was no real person being punished/compensated, and suggested that the only way paying to alter the payoffs of a fictional character wasn’t a total waste is if participants were trying to signal to the experimenter, in which case maybe we need to think about the participant-experimenter dynamic. I’m fine with the authors’ response regarding anonymity, but I do need to raise another concern with a fake Player A: From the perspective of the female rater, why shouldn’t we expect her to think that men who just give away their money to punish or compensate non-existent characters aren’t the best partners? Shouldn’t “do nothing” be the most attractive option, at least for a subset of women who see through the fiction? (Would presumably apply in the broader social selection framework too – do we want an ally who can’t distinguish reality from fiction?) I don’t think that another study with real stakes is required, but this concern should be discussed. (And it isn’t totally clear that the women knew that the men were punishing/compensating a fake Player A. If they didn’t know, this concern disappears, although then we might need to discuss the use of deception, unless the hypothetical-ness was clear.)

4.

“Women preferred strength for long-term relationships... Women preferred attractive men for short-term relationships” – I don’t understand what these claims are based on. Women rated hypothetical men’s strength and attractiveness as a function of the men’s decision and the women’s relationship context. So we can know which male actions women prefer, but how do we know which traits they prefer in different contexts? Aren’t the traits entirely based on the actions?

5.

In general, since 3.3.4 suggests that all of the ratings load to some “mate quality” factor, and since “strength” seems to have a fuzzy meaning as deployed here, it is worth trying to parse the meanings of each individual rating word?

But if we want to do that, it seems to me that punishment could signal literal physical strength when the mode of punishment involves physical action (e.g., detaining and/or assaulting the perpetrator), whereas the mode of (fake) punishment here was economic. The first part in the general discussion about “strength” (line 682-92) was entirely based on a metaphorical notion of the word – ignoring the primary (somatic) definition of the word seems odd.

6.

Table 1 and related text: Could punishment have long-term reputation costs, e.g. a reputation for violence or cruelty? (And, as mentioned previously, could punishment or compensation of fictional characters have reputations costs?)

7.

From footnote 1, it sounds like the Study 1 “active control” condition participants were run months after the others. I’d suggest stating that explicitly rather than leaving it implicit.

8.

I don’t object to the content of section 4.2, but it seems unnecessary in this paper, given that the methods were clearly aimed at mating psychology, rather than a more general cooperative psychology. Acknowledging that sexual selection can be seen as a subset of social selection (or perhaps other interpretations) and crediting important work on this topic could be accomplished far more efficiently, probably in the Intro.

9.

I suspect that the paper would read more smoothly if a lot of the technical details of 3.3.4 and maybe 3.3.5 were relegated to Supplement, with only summaries of that stuff in the main text.

10.

Figure text is difficult to read. Could be the journal system’s auto-PDF distorting it?

Author's Response to Decision Letter for (RSOS-181441.R2)

See Appendix D.

RSOS-181441.R3 (Revision)

Review form: Reviewer 3

Is the manuscript scientifically sound in its present form?

Yes

Are the interpretations and conclusions justified by the results?

Yes

Is the language acceptable?

Yes

Do you have any ethical concerns with this paper?

No

Have you any concerns about statistical analyses in this paper?

No

Recommendation?

Accept with minor revision (please list in comments)

Comments to the Author(s)

The authors have adequately addressed concerns I raised in previous reviews. I thank them for a nice paper!

I still notice some errors -- need a bit more proofing. Aside from that, I recommend accepting the manuscript.

Cheers!

Decision letter (RSOS-181441.R3)

08-Aug-2019

Dear Dr Ferguson,

I am pleased to inform you that your manuscript entitled "To Help or Punish in the Face of Unfairness: Men and Women Prefer Mutually-Beneficial Strategies over Punishment in a Sexual Selection Context" is now accepted for publication in Royal Society Open Science.

Kind regards,

on behalf of Dr Antonia Hamilton (Subject Editor)
openscience@royalsociety.org

Reviewer comments to Author:

Reviewer: 3

Comments to the Author(s)

The authors have adequately addressed concerns I raised in previous reviews. I thank them for a nice paper!

I still notice some errors -- need a bit more proofing. Aside from that, I recommend accepting the manuscript.

Cheers!

Appendix A

The University of
Nottingham

UNITED KINGDOM • CHINA • MALAYSIA

Faculty of Science

School of Psychology

The University of Nottingham
University Park
Nottingham
NG7 2RD

t: +44 (0)115 951 5361

f: +44 (0)115 951 5324

e: psychology-enquiries@nottingham.ac.uk

www.nottingham.ac.uk/psychology

29th August 2018

Re: RSOS-180216 entitled "Compassion Wins Out: Sexual Selection and Men and Women's Preferences for Punishment and Compensation"

Dear Dr. Crockett and Dr. Hamilton,

Thank you for the opportunity to revise and resubmit our paper (RSOS-180216 "Compassion Wins Out: Sexual Selection and Men and Women's Preferences for Punishment and Compensation"). This was originally submitted to *Biology Letters* (RSBL-2017-0532.R1a) and transferred to RSOS and that is why the introduction, results and discussion lacked detail as they were formatted for the word constraints of *Biology Letters*. Therefore, we very much welcome the opportunity to expand on these sections to make our case for the novelty of the work more clearly. We have also added a third experiment to replicate and extend the results of experiment two using a mixed between-within design. Thank you for your helpful comments and suggestions and we detail below how we have attempted to deal with each of these. I have also included the reviewers' comments from the original submission to *Biology Letters* and my response to these. This revised version addresses all the issues raised by the original reviewers and we detail some additional responses at the end of this response.

Responses to RSOS Editors comments (Dr. Crockett and Dr. Hamilton)

Issue 1.

First, I expected to see a more comprehensive literature review and depth of scholarship. The introduction is just over 300 words, with 12 references. As there is an extensive literature on sexual selection and punishment, much more needs to be done to motivate the current study in the context of prior relevant literature. A similar criticism applies to the discussion, at less than 400 words and 8 references.

Response 1.

I have now extended the introduction to just over 7 pages, 61 references and 2,249 words (line 34 to 206). The introduction now contains an over view of the central focus of the paper, a review of punishment, compensation and compassion in relation to sexual selection, the role of traits, political ideology and emotions, and assessment beyond attraction. I have now extended the discussion to 5 pages and 1,384 words (Lines 552 to 658). The discussion contains an extended examination of the implications of the results, future directions and limitations. The shorter introduction and discussion in the original submission was due entirely to the paper having been formatted for *Biology Letters* where the final decision was not to accept it for *Biology Letters* but to transfer to *RSOS*. Hence, the word constraints of *Biology Letters* carried over. We have really welcomed the chance to extend the introduction and discussion as this allows us to expand on the case for the novelty of the work and how we feel it adds to the literature on sexual selection of prosociality, specifically by exploring restorative justice preferences including punishment.

Issue 2.

Second, there are issues with the methodology. The sample size for Experiment 1 is not well justified. The experimental conditions in Experiment 2 are not counterbalanced, which strongly limits the conclusions that can be drawn. I also have concerns that a (non-counterbalanced) within-subjects 2 x 4 design may alert participants to the hypotheses of the study, and it is not clear whether anything was done to counter possible demand effects. I

would also recommend double-checking the reported statistics as there seem to be some implausible values (e.g. on pg 7 line 142).

Response 2.

We have now provided sample size justifications are provided for each experiment (experiment one lines 217 to 222, experiment two lines 289 to 293 and experiment three lines 314 to 318). We have also conducted a third experiment using a mixed between (long-term vs short term) by within (restorative justice preferences) design. It is powered based on experiment two and replicates the effects observed in experiment two, and extends these to include additional ratings of the man's compassion, fairness and strength. These additional ratings, especially fairness and compassion, allows us to be more certain in our conclusions that compensatory strategies signal compassion, but also add to this by showing that they are also signal fairness. We also included measures of the female rated empathic concern and political ideology to explore conjectures suggested by one of the original reviewers. The assessment of empathic concern allows for some direct compare the results of experiment there to experiment one and suggests how empathic concern may be a correlated sexually selected trait. All statistics have been double checked.

Issue 3

Finally, given how short the paper is, it was unclear why many of the analyses were placed in supplementary materials, including analyses designed to test central hypotheses around the role of compassion.

Response 3

Again the large amount of materials in the supplementary files was a consequence of the word limits of the original *Biology Letters* submission. We welcome the chance to move some of these across. I have now added a multinomial regression to the main paper to describe the effects of empathic traits and emotions on restorative justice preferences for experiment one (lines 374-394). I have also included the main figure of preference for study 1 (line 371). The supplementary files now primarily contain materials and robustness checks.

Issue 4

I would be willing to reconsider a new submission that addresses the above concerns, however it would need to include a much more comprehensive literature review and discussion that describes in more detail previous work on this topic, providing context for the new studies, a more comprehensive results section, and a replication of the key findings with a fully counterbalanced and ideally between-subjects design, before I can justify sending this out for further peer review.

Response 4

Given our responses to issues 1 to 3 above, we have now expanded and developed a more comprehensive literature review and discussion, made the results section much more comprehensive and detailed and replicated the findings from experiment two with a mixed between-within design in experiment three.

Given these changes, additional analyses and an additional experiment we hope you feel that the paper is strong enough to go out for external review.

I can confirm that all authors have read and agreed the final manuscript.

I look forward to hearing from you.

Yours sincerely

Eamonn Ferguson PhD (on behalf of all co-authors)

Original Reviewer's Comments (Biology Letters) and my Response to them

School of Psychology
University Park
Nottingham
NG7 2RD
Tel +44 (0)115 9515284
Fax +44 (0)115 9515324
Head of School:
Professor Paul McGraw

3rd November 2017

RE: RSBL-2017-0532. "Compassion Wins Out: Sexual Selection and Men and Women's Preferences for Punishment and Compensation".

Dear Professor Bennett,

Thank you for organizing the review of our paper "Compassion Wins Out: Sexual Selection and Men and Women's Preferences for Punishment and Compensation" for *Biology Letters* and for both your and the reviewers' very helpful and insightful comments. We have addressed all the comments and our responses are detailed below and highlighted in yellow in the text. The main changes we have added are:

1. Re-worked the introduction to frame the paper more clearly around the tension between punishment and compensation in a restorative justice context.
2. Added an additional control group, to examine if the effects we report are just attributable to priming, rather than the content of the prime. We find that the effect we reported originally stand and were not attributable to just being primed.
3. We add caveats concerning the samples and simple sizes to the discussion.
4. We have re-worked the discussion.

The paper is 2, 652 words and thus 152 words over the word limit. However, I believe we are allowed to go over the limit by about 200 words in order to ensure that we address the reviewers' comments. I hope that this is OK.

Again, we thank you and the reviewers for your time and comments – which we feel have very much helped to strengthen the paper – and hopefully we have addressed these fully.

Comment from the Handling Editor

It has been proposed by the reviewers that the idea needs to come across more clearly that in spite of the fact that costly punishment can be conceived of as a form of altruism, it may also

represent a special case which does not necessarily increase mate attractiveness. I think you need to cover this alternative aspect in order to accommodate some of the concerns of reviewer 1 regarding the novel contribution of the study. Reviewer 1 also notes the lack of a control prime condition, which would certainly be a valuable addition to the manuscript. Both of the reviewers raise concerns about the extent to which these results might generalise across cultures (or even alternative populations within a culture), which also needs to be taken seriously into consideration. Please revise your paper in light of the points raised in the review process. I would appreciate a detailed covering letter highlighting how you have taken on board the points raised in the review process.

Responses to Handling Editor

Issue 1

It has been proposed by the reviewers that the idea needs to come across more clearly that in spite of the fact that costly punishment can be conceived of as a form of altruism, it may also represent a special case which does not necessarily increase mate attractiveness. I think you need to cover this alternative aspect in order to accommodate some of the concerns of reviewer 1 regarding the novel contribution of the study.

Response 1

We thank the reviewer's for these comments as they have helped us to, hopefully, frame the introduction more clearly to highlight the novelty of the study. We feel the novelty of our study lies in how men resolve the tension between punishing and compensating in a **restorative justice** context, which has particular implication for sexual selection theory. While previous studies by *Iredale et al.*, *Tognetti et al.*, or *Griskevicius et al.*, for example, have, indeed, shown that under romantic conditions (aromatic-primed or presence of a female) men are more **generous** than in a control-prime or on their own. However, they did not explore how people display pro-sociality when an unfairness is witnessed. That is, what is the preferred method to restore justice and what does this signal? **Generosity** in studies by *Iredale et al.*, *Tognetti et al.*, or *Griskevicius et al.*, for example, is operationalized in terms of financial donations to a charity, or to the public good, or display of conspicuous consumption. While such acts signal generosity and pro-sociality other attributes linked to signalling pro-sociality (e.g., compassion for victims, fairness norm enforcement) can also be signalled by restorative justice options. The novelty of our study lies in exploring how men resolve the tension between compensation and punishment when in a romantically primed context. We argue that this is important from an evolutionary sexual selection perspective as both punishment and compensation are viewed positively (Raihani & Bshary, 2015), but for potentially different reasons. Punishment signals trustworthiness (Jordan et al., 2016) and compensation signals compassion (see Zhao et al., 2017). Furthermore, punishment is transgressor focused while compensation is victim focused. Punishment may be motivated by a punitive motives, such as *just-deserts*, along with a desire to enforce fairness norms and make for a safer society. On the other hand compensation is focused on community values and compassion. Thus, both are likely to be attractive for different reasons. Therefore, the novel question we ask, for the first time, is which is preferred by men in a romantic context – punishment or compensation? We also ask, for the first time, which strategy – punishment or compensation – do women prefer? Thus, we develop on the existing literature beyond its current state, which has examined simple acts of generosity, to the restorative justice context

and how pro-sociality is signalled by men when faced with punishment vs compensation options, and which strategy women prefer.

The papers by Barclay (2010) and Arnocky et al. (2017) focus on how generosity influences attractiveness and mating success. These papers indeed, highlight that generosity is attractive. They, however, do not explore other strategies that may be used to display pro-sociality - punishment and compensation – and if these displays enhance attractiveness or vary across romantic and non-romantic contexts.

Based on the above we have completely re-worked the introduction (p 3 para 1) to address these points and, hopefully, show why we think both punishment and compensation should be attractive to women and why we feel this is a novel contribution to the literature.

Issue 2

Reviewer 1 also notes the lack of a control prime condition, which would certainly be a valuable addition to the manuscript.

Response2

This is a valuable and important point and we have now added an additional control group to explore any potential effects attributable to being primed rather than the content of the prime (p4. Para 1 line 54 onwards and ‘Supplementary File S1’). In our original design we randomly allocated participants to either a ‘romantic-prime’ and a ‘pure-control (no prime)’ condition. Our rationale for this was based on findings reported by Griskevicius et al. These authors showed that when a ‘romantic-prime’ is contrasted with a ‘control-prime’ (i.e., describing the preferred weather, while look at building on an ordinary street) a significant effect for the ‘romantic-prime’ emerges on generosity. Thus, we felt that the effects observed for the romantic-prime were not a consequence of being primed, but depended on the type of prime. We also wanted a control that resembled as near as possible the standard 3PPC game, so we opted for the pure-control vs the romantic-prime in our original design. However, the reviewer raises an important point so we have collected additional data (N = 23 men) using the Griskevicius et al. control-prime procedure. This involves participants choosing one of three ordinary street scenes and spending up to 3 minutes describing the ideal weather they would like to experience while looking around the buildings on the street they choose. We now include this in the supplementary files (Supplementary File S1) and mention it in the paper (p4. Para 1 line 54 onwards). In our original experiment we randomly allocated people to one of two conditions (romantic-prime or pure-control), however the ‘control-prime’ data represent an additional condition that was not part of our original randomization. As such, the ‘control-prime’ data lies outside our randomization procedure and means we cannot infer causality if we include this in our main design and analysis. Therefore, we include it in the Supplementary Files (S1) as additional information requested by reviewers. The results show that, in fact, the pattern of results observed in the ‘control-prime’ is identical to those observed in the ‘pure-control’ and that the effect we present contrasting the pure-control with the romantic-prime replicate when we contrast the new ‘control-prime; with the ‘romantic-prime’. Thus, we feel that this strengthens our overall conclusions without losing the power of our initial randomization. We feel this additional data strengthens the paper and we thank the reviewer for raising it.

Issue 3

Both of the reviewers raise concerns about the extent to which these results might generalise across cultures (or even alternative populations within a culture), which also needs to be taken seriously into consideration

Response 3

We thank both reviewers for raising this and we have now added this to our discussion (p 8, para 2). We state how our results may reflect the use of a WEIRD sample and go on to suggest that indeed we may observe different results in different cultures.

Referee 1

This article addressed two main questions: (1) are males likely to punish transgressors and/or compensate victims, to signal their altruistic nature, after receiving a mating prime? And (2) are the strategies employed by males used in female mate choice? I found these questions interesting and was excited to read the manuscript, especially since the authors focus on both male behaviour and female perceptions (it is unusual but commendable for papers to examine both sides). The main results are that (1) reading a mating prime made men more likely to compensate victims of inequality (relative to no prime), but had no effect on men's tendency to punish; and (2) women were attracted to men who were nice (i.e., compensated victims) but not to men who punish. Most of the results correspond to some previous research on the effects of mating on altruism (and vice versa). However, I must admit that I was somewhat disappointed with the execution of the studies, as well as the writing of the manuscript. I have major concerns regarding both of these issues, which I explain below. On a positive note, I think they are good ideas with a strong theoretical foundation.

We thank the reviewer for their comments. We have, hopefully, addressed all the major concerns below.

Issue 1

What do these results add that is not already present in previous research such as Iredale et al., Tognetti et al., or Griskevicius et al. showing that romantically-primed men are more generous, or Barclay (2010, Br. J. Psych, v101, pp123-135) or Arnocky et al. (2017 Br. J. Psych, 108, 416-435) showing that women are attracted to men who are altruistic? Compensation seems very much the same as altruism or generosity, which have both been studied many times. The novelty of the current manuscript needs to be clarified. If the punishment question is new, then bring that out more.

Response 1

We thank the reviewer's for these comments as they have helped us to, hopefully, frame the introduction more clearly to highlight the novelty of the study. We feel the novelty of our study lies in how men resolve the tension between punishing and compensating in a **restorative justice** context, which has particular implication for sexual selection theory. While previous studies by *Iredale et al., Tognetti et al., or Griskevicius et al* , for example, have, indeed, shown that under romantic conditions (aromatic-primed or presence of a female) men are more **generous** than in a control-prime or on their own. However, they did not explore how people display pro-sociality when an unfairness is witnessed. That is, what is

the preferred method to restore justice and what does this signal? **Generosity** in studies by *Iredale et al.*, *Tognetti et al.*, or *Griskevicius et al.*, for example, is operationalized in terms of financial donations to a charity, or to the public good, or display of conspicuous consumption. While such acts signal generosity and pro-sociality other attributes linked to signalling pro-sociality (e.g., compassion for victims, fairness norm enforcement) can also be signalled by restorative justice options. The novelty of our study lies in exploring how men resolve the tension between compensation and punishment when in a romantically primed context. We argue that this is important from an evolutionary sexual selection perspective as both punishment and compensation are viewed positively (Raihani & Bshary, 2015), but for potentially different reasons. Punishment signals trustworthiness (Jordan et al., 2016) and compensation signals compassion (see Zhao et al., 2017). Furthermore, punishment is transgressor focused while compensation is victim focused. Punishment may be motivated by a punitive motives, such as *just-deserts*, along with a desire to enforce fairness norms and make for a safer society. On the other hand compensation is focused on community values and compassion. Thus, both are likely to be attractive for different reasons. Therefore, the novel question we ask, for the first time, is which is preferred by men in a romantic context – punishment or compensation? We also ask, for the first time, which strategy – punishment or compensation – do women prefer? Thus, we develop on the existing literature beyond its current state, which has examined simple acts of generosity, to the restorative justice context and how pro-sociality is signalled by men when faced with punishment vs compensation options, and which strategy women prefer.

The papers by Barclay (2010) and Arnocky et al. (2017) focus on how generosity influences attractiveness and mating success. These papers indeed, highlight that generosity is attractive. They, however, do not explore other strategies that may be used to display pro-sociality - punishment and compensation – and if these displays enhance attractiveness or vary across romantic and non-romantic contexts.

Based on the above we have completely re-worked the introduction (p 3 para 1) to address these points and, hopefully, show why we think both punishment and compensation should be attractive to women and why we feel this is a novel contribution to the literature.

Issue 2.

b) Would the authors expect the same results in different societies where toughness or fierceness is valued, such as the Yanomamo? Might women have preferred punishers in such a society? This warrants discussion.

Response 2

We thank both reviewers for raising this and we have now added this to our discussion (p 8, para 2). We state how our results may reflect the use of a WEIRD sample and go on to suggest that indeed we may observe different results in different cultures.

Issue 3

For experiment 1, the control group did not complete a priming task. Because of this, one could argue that the results obtained were simply because they completed a priming task, not because of the content of the priming task. A similar (but control) task should be used for the control condition, such that ideally there are three conditions: romantic prime, neutral prime, no prime.

Response 3:

This is a valuable and important point and we have now added an additional control group to explore any potential effects attributable to being primed rather than the content of the prime (p4. Para 1 line 54 onwards and 'Supplementary File S1'). In our original design we randomly allocated participants to either a 'romantic-prime' and a 'pure-control (no prime)' condition. Our rationale for this was based on findings reported by Griskevicius et al. These authors showed that when a 'romantic-prime' is contrasted with a 'control-prime' (i.e., describing the preferred weather, while look at building on an ordinary street) a significant effect for the 'romantic-prime' emerges on generosity. Thus, we felt that the effects observed for the romantic-prime were not a consequence of being primed, but depended on the type of prime. We also wanted a control that resembled as near as possible the standard 3PPC game, so we opted for the pure-control vs the romantic-prime in our original design. However, the reviewer raises an important point so we have collected additional data (N = 23 men) using the Griskevicius et al. control-prime procedure. This involves participants choosing one of three ordinary street scenes and spending up to 3 minutes describing the ideal weather they would like to experience while looking around the buildings on the street they choose. We now include this in the supplementary files (Supplementary File S1) and mention it in the paper (p4. Para 1 line 54 onwards). In our original experiment we randomly allocated people to one of two conditions (romantic-prime or pure-control), however the 'control-prime' data represent an additional condition that was not part of our original randomization. As such, the 'control-prime' data lies outside of randomization procedure and means we cannot infer causality if we include this in our main design and analysis. Therefore, we include it in the Supplementary Files (S1) as additional information requested by reviewers. The results show that, in fact, the pattern of results observed in the 'control-prime' is identical to those observed in the 'pure-control' and that the effect we present contrasting the pure-control with the romantic-prime replicate when we contrast the new 'control-prime' with the 'romantic-prime'. Thus, we feel that this strengthens our overall conclusions without losing the power of our initial randomization. We feel this additional data strengthens the paper and we thank the reviewer for raising it.

Issue 4.

The sample size is relatively small in study 1. Although the > 5 rule is just a rule of thumb, it is ideal to have obtained n 's to be larger than that. With small sample sizes, there is greater uncertainty about the magnitude of the effect size. Furthermore, the rate of false findings is lower as statistical power increases, i.e., given that you found something, what is the probability that your finding reflects a false positive instead of a real effect? This is different from p -values and α ... this is $\alpha / (\alpha + \text{power})$. The authors did find a large effect of priming on compensation, which makes me feel OK about that, but the study is underpowered to detect any effects (or non-effects) of punishment. A higher powered study should be conducted if the authors wish to say that mating primes don't affect men's punishment, in order to show that the lack of effect is not a false negative.

Response 4.

We thank the reviewer for these helpful comments. We have collected extra data on a 'control-prime' that replicates virtually exactly the effects observed for the pure-control and increase the N and power. Thus, we feel that this adds some evidence for the robustness of

these findings. However, we also added a caveat about the sample size in the discussion (P 8, para 2).

Issue 5.

The authors do not explain why they used certain statistics, which made the presentation of their results confusing.

Response 5.

We apologize for this lack of clarity and we have now added additional explanation (p 6, para 2, L 100) and moved the additional statistical from the main text to the supplementary files so that we can focus more on explaining the results and analyses more clearly (see Supplementary files S1, S3, S4, S5)

Issue 6.

Effect sizes should be reported for all analyses.

Response 6.

We now report these. We report Phi for Chi-Square, partial eta-square for the ANOVAs and Cohen's d for the t-tests.

Issue 7.

The results of Study 1 basically show that priming affects men's compensation, but not punishment: there is no overall change in punishment (the drop in "Punish" is compensated by an increase in "Both", suggesting that unprimed men who would have only punished will tend to both punish and compensate after priming). Can the authors do some statistics to explicitly show this effect? They combine groups to show a main effect of priming on compensation... they should do something similar with punishment.

Response 7.

There were nine men who punished in the pure-control and one in the romantic-prime. Conversely, two men both compensate and punish in the pure-control and ten who both compensated and punished when romantically-primed. Thus, eleven men in the pure-control either punished or both compensated and punished and eleven men in the romantic-prime either punished or both compensated and punished. Thus, we see that the amount of punishment remains the same across the two conditions it is how this is supplemented by compensation that changes. We now present this idea in the discussion, specifically that men when romantically-primed by switching to either compensate or supplement their punishment with compensation (p 8, para 1, 149-151).

Issue 8.

*Also, the abstract (and intro to a certain extent) is written as though punishment *decreases* under the mating priming, where in reality it doesn't decrease... the overall punishment stays virtually unchanged. This needs to be rewritten.*

Response 8.

This has been re-written the abstract to better reflect the pattern of results observed,

Issue 9.

The abstract says that women were least attracted to men who punished. The results do not reflect this. Women were equally attracted to men who punished and men who did nothing. (Punishment had a negative effect on the attractiveness of men who also compensated, but that's slightly different.) This phrasing must be written to be more accurate, for example by saying that punishment did not increase men's attractiveness and that it reduced the attractiveness of compensators.

Response 9.

We have now re-written this to say that women see men who punish and do nothing as equally attractive, but less attractive than compensator and those who compensate and punish (p 8, para 1, 146-149).

Issue 10.

The paper needs better organization, as it has many non-sequiturs, grammatical errors, and needs to be proof-read.

Response 10.

We apologize for this. The paper has now been proof read. We hope that the structure, grammar and flow is now better.

Issue 11.

b. For organization, it would be much clearer if each experiment was presented on its own, with background, methods, then results, rather than both methods and then both results. This saves the reader from jumping back and forth.

Response 11.

We thank the reviewer for this suggestion. We feel the structure we have – which has been used in Biology Letter before is an expedient way to present the methods and results. As such, we have stuck with this format but tried to make the comprehensibility greater by referring more clearly to experiment one and experiment two in both the method and results. However, if the reviewer still feels that this is confusing we will endeavour to re-write as two sequential studies.

Issue 12.

The authors seem to have a good understanding of evolutionary theory and mate preferences, however, they do not always communicate their ideas clearly. Many of the concepts need clarification because they make the assumption that the readers are already familiar with the content (e.g., altruism is undefined despite very different usages to different authors, signaling, what the specific norms are). As such, the framing, setup of research questions, and presentation of results needs elaboration.

Response 12.

We have now clarified and defined all terms in the paper as suggested. For example we have completely re-worked the introduction and now talk about pro-sociality – now defined on page 3 (para 1, lines 22-23)

Issue 13.

Methods and results are unclear to non-experts (e.g., Dictator Game not explained, what happened to data from the 15 non-heterosexual women, introducing compensation and punishment on line 63 could be more explicit that compensation increases B' earnings and punishment reduces A's earnings). I do not know why some decisions were made (e.g., why choose particular statistical tests, what groups are being compared in each test). Each decision needs to be made more explicit.

Response 13.

We thank the review for these comments. We have used the phrasing suggested by the reviewer (p 4 para 2 line 68-70) we also justify why we focus on heterosexual women in experiment two and that other sexualities were excluded post data collection (p 5, para 2, ls 82-85).

Issue 14.

Some of the reported statistics are also mixed up (reported stats for wrong groups, they say Fisher exact test but then present Chi-Square).

Response 14.

We have ensured that we state our analyses clearer. For example we have now used the Yate's correct for the low sample sizes for the Chi-Square, which is more appropriate than the Fisher exact as none of the expected cell frequencies is below 5 (p 6, para 2, lines 100-102). All the additional statistics that were not fully described in the original main paper have now been moved to supplementary files (see Supplementary files S1, S3, S4, S5) and described in detail there.

Issue 15

Some statistics are unclear (e.g., does “minimum p” mean “all p's greater than”?, what's the one-sample t-test compared against) g. The discussion is a bit thin.

Response 15.

We have removed all these non-standard phrases and moved all these additional statistics, that were not fully described in the original main paper, to supplementary files (see Supplementary files S1, S3, S4, S5) and described in full detail there. We also now state that the one-sample t-test is compared to the minimum value on the scale of one as this indicate completely fair (p 7, para 2, line 124). However, the difference is significant is you compare to the mid-point of 3.5 ($M = 4.70$, $SD = 1.50$; $t(103)_{\text{one sample}} = 8.17$, $p = .000$; Cohen's $d = 0.80$).

Issue 16.

Some references are off. For example, Wood & Brumbaugh is cited as saying that “women prefer altruism in a mate”. That article seems to show nothing of the sort... the closest is “soft-hearted”. If the authors want to say that women prefer altruism, then they should cite an article that actually tests that.

Response 16.

We have revised all the references to make them more appropriate.

Referee 2

This is a good, clear, strong paper that reports two innovative studies that address a key issue in human evolution: the possible emergence of kindness, altruism, and generosity through sexual selection.

The experiments are well-designed and well-run, presented clearly, with compelling results. Although the sample sizes (60 men in study 1, 119 women in study 2) are somewhat small compared to survey studies, study 1 entailed individual testing in an experimental lab, using a behavioral economics paradigm, which would have been a lot of work. Study 2 is a perfectly adequate sample of purposes of rating the male behaviors.

Response

We thank the reviewer for these supportive comments.

Issue 1.

The tension in these studies between altruism (compensating the 'victim' of an unfair exchange) and punishment (of the unfair person) is interesting, and the results are a little surprising. I thought the female raters might be a bit more attracted to punishing males, given the common attraction to superhero vigilantes, cops, soldiers, etc, and their frequent appearance in romance novels and fan fiction. But then, there is the higher attraction to punishment for short-term mating. Maybe comment a bit more on that in the discussion.

Response 1.

We have highlighted this tension and made it much more of the re-worked introduction (p 3 para 1). We also highlight in the discussion the fact that punishment is indeed a higher preferred strategy for short-term relationships and suggest this might reflect a preference women show for cad's over dad's for short-term relationships (P 8, para 1, Ls 151-157) . We

also state how our results may reflect the use of a WEIRD sample and go on to suggest that indeed we may observe different results in different cultures.

Issue 2.

In section 2.0 methods, it would be helpful to identify the 'UK university' as U. Nottingham if that's where the research happened. I think it's always good to be more specific about the participants and where they live. It would also be helpful to give readers a sense of how selective Nottingham is (as an cue of participant intelligence and conscientiousness), and their racial/ethnic mix (if available -- otherwise just give a sense of what the overall Nottingham demography is like).

Response 2.

We now provide much more detail in the sample in Supplementary file3 (see main text p 6, para 1 lines 95-97) and this file also contain data on the Nottingham University.

Issue 3.

In terms of male and female behavior and mate preferences, it would also be extremely useful to get a sense of the participants' political attitudes -- insofar as liberals tend to be more attraction to compassion and conservatives to punishment. Are these participants mostly quite politically liberal? Or at least are Nottingham people in general like that? Maybe note the results could be different with more conservative samples? (And/or older adults?)

Response 3.

We thank the reviewer for this comment. We present detail on the political make-up of the students at Nottingham University. We do not have detail on this for our sample but now at least are able to give the wide context (supplementary file S2). However, we do suggest that future work may want to consider political valise and beliefs (p 8, para 2).

Issue 4.

Figure 2 is confusing -- maybe draw a vertical line between the four bars on left and four on right, and label the left four 'short-term mating' and the right four 'long-term mating', if I have those the right way around.

Response 4.

We have now added this. Thank you for the comment

Overall, a very nice paper, well worth publishing in this journal.

Again thank you for your supportive comments and feedback.

References

Jordan JJ, Hoffman M, Hoffman, P, Rand DG. 2016. Third-party punishment as a costly signal of trustworthiness. *Nature*. **530**, 473-477. (doi:10.1038/nature16981)

Raihani NJ, Bshary, R. 2015. Third-party punishers are rewarded, but third-party helpers even more so. *Evol*. **69**, 993-1003. (doi:10.1111/evo.12637).

Zhao K, Ferguson E, Smillie L. 2017 Politeness and Compassion Differentially Predict Adherence to Fairness Norms and Interventions to Norm Violation. *Sci Reps*. **7**, 3415. (doi: 10.1038/s41598-017=02952).

We hope that the changes we have made to the paper have improved its quality and readability.

Yours sincerely,

Eamonn Ferguson.

Response to Editorial Decision on the Revised Paper

School of Psychology
University Park
Nottingham
NG7 2RD
Tel +44 (0)115 9515284
Fax +44 (0)115 9515324
Head of School:
Professor Paul McGraw

4th December 2017

Appeal to Reject Decision for RSBL-2017-0532.R1 entitled "Compassion Wins Out: Sexual Selection and Men and Women's Preferences for Punishment and Compensation"

Dear Professor Bennet,

Thank you for organizing the review process and to the reviewer for their comments for our resubmitted manuscript "Compassion Wins Out: Sexual Selection and Men and Women's Preferences for Punishment and Compensation". While we fully understand the decision to reject based on the reviewer's comments we feel that the opinions presented by the reviewer reflect a misunderstood the nature of the previous work, and as such their interpretation of our studies novelty. We were perhaps were not as explicit in explicating the novelty as we had thought. We show more explicitly below how reviewer 1's comments reflect a misunderstanding of previous literature and why our study adds new insights. Indeed, reviewer 2, who did not comment on the revised paper, is a co-author on the Griskevicius et al study that reviewer 1 claims we are essentially replicating. Reviewer 2 did not indicate that what we were doing was a replication of their study, in fact they commented on its novelty stating: *"This is a good, clear, strong paper that reports two innovative studies that address a key issue in human evolution: the possible emergence of kindness, altruism, and generosity through sexual selection"* and go on to state: *"The tension in these studies between altruism (compensating the 'victim' of an unfair exchange) and punishment (of the unfair person) is interesting, and the results are a little surprising.*

We detail below why we have concerns with the reviewer argument on the novelty of our study and also detail how we can address all the minor comments raised by the reviewer.

In addition, given we made a number of major changes (included new data for a new experimental condition, re-wrote the introduction and addressed many commitments by the second reviewer reviewer), we were concerned that reviewer 2 was not given the opportunity to comment on the new draft (and potentially the concerns raised by reviewer 1 in the initial review).

Therefore, given we feel there are some inaccuracies in the single reviewer's reading of the literature we would like to submit and appeal to the decision to reject.

Reviewer 1.

Issue 1

*“My big comment is still that it is not clear how big an advancement this paper represents over previous work. It is currently framed about testing whether “restorative justice” is desirable... why is it important to test this? Again, it seems like compensation is marginally different from generosity. For example: suppose that A observes B suffer a loss or fail to receive something, and A decides to help. Why does it matter whether B's loss is caused by bad luck (e.g., failed hunting, randomly assigned to the Receiver role in an experiment), natural disasters, predators, or another person C who acted unfairly? It is the same trait that is being signaled in all cases (i.e., willingness to help B). I am not convinced that compensation is fundamentally different from generosity; if anything, compensation is a subset of the larger category of generosity. So while the results may be publishable *somewhere* with additional revisions & clarifications (see below), it is a question of where, and whether it is interesting enough for the general audience of Biology Letters. I might recommend somewhere that accepts replication experiments.”*

Response:

There are number of points here to address with the reviewer's argument that our study does not offer anything new.

“Again, it seems like compensation is marginally different from generosity. For example: suppose that A observes B suffer a loss or fail to receive something, and A decides to help. Why does it matter whether B’s loss is caused by bad luck (e.g., failed hunting, randomly assigned to the Receiver role in an experiment), natural disasters, predators, or another person C who acted unfairly? It is the same trait that is being signaled in all cases (i.e., willingness to help B).”

This example given by the reviewer does not show that these acts are fundamentally the same as they have different motivational and attributional structures. The attributional processes that would underlay the decisions here are very different. Bad luck is something that happens by *chance* for example (being in the wrong place at the wrong time) and in this case, one may feel sorry for the person and help them as they are not responsible for their fate. The same is true for ‘*natural disasters, predators*’ etc. This is *not* the case when someone (person C) actively decides to act unfairly towards someone else. In this case, the act of person C is deliberate and the victim’s loss is not bad luck but due to an active decision on the behalf person C. In this case, one may want to help the victim but also may want to punish person C. I cannot do anything about a natural disaster or bad luck other than help the victim (there is no one to punish or appeal to). In the cases of the *natural disaster* or *bad luck* there are only two choices in the choice set (do nothing or help the victim). When someone has intentionally hurt someone there are 4 choices (do nothing, help the victim, punish the perpetrator or both punish the perpetrator *and* help the victim). Thus, the choices facing participants in our studies are very different. Indeed a large body of evidence in economics and psychology shows that changing the choice set dramatically alter how people respond. The studies by Griskevicius, Tognetti, Iredale, and others only look at the first case with 2 choices (do nothing or help). However, in the case of active transgression there are 4 choices and in this context is not clear from the existing literature what romantically primed men would do. It has never been tested before and is fundamentally different to the choice set faced in the studies by Griskevicius, Tognetti, Iredale, and others. Indeed, reviewer 2 in the original round of reviews highlighted this as the novel aspect of the study. Furthermore, reviewer 2 is a co-authors on the Griskevicius et al study and did not indicate that what we were doing as a replication of their study, in fact they commented on its novelty. We feel that the reviewer’s argument is conflating the 2-choice and 4-choice sets. This is why they see the

study as not adding anything new. We may have not made this clear enough and we can make this much more explicit, by reference to choice sets as detailed above.

The reviewers initial (“*It is currently framed about testing whether “restorative justice” is desirable... why is it important to test this?*”) is based on the above inaccuracy and therefore is not justified. Examining restorative justice is important as it offers the 4-choice set (do nothing, help the victim, punish the perpetrator or both punish and help) that is open to people when they see someone else being *actively unfairly* treated. We are the first to explore what men do and what women want in this ecologically valid context. From an evolutionary perspective, when faced with this situation, men could show their ‘empathic side’ and compensate the victim or could demonstrate their dominance by punishing the perpetrator. Both have implications for the evolution of pro-sociality via sexual selection and have not been examined before. Study 1 is thus not a just a replication of Griskevicius, Tognetti, or Iredale as we have a 4-choice restorative justice scenario and they have a 2-choice helping only scenario. Study 2 is not just a replication of Barclay, Arnocky, Farrelly as in those studies women were not presented with the 4 choice set just 2 (men who were altruistic or not). Indeed, in the original reviews, reviewer 2 predicted that in our 4-choice set that men would punish (to demonstrate dominance) and found our results to be unexpected and intriguing.

Thus, we feel that the reviewer may have misunderstood the nature of the study in the context of the existing literature and we apologise if we had not made our distinction clear enough.

Issue 2

It is more interesting to look at punishment, which is more novel, but it is also unclear that these laboratory situations would be enough to trigger a desire for a punisher. One-shot games are ambiguous about players’ motives (hence “mixed-motive games”), so it is hard to tell the difference between someone who is punishing for a good reason (“restorative justice”) and someone who is punishing because they like to hurt people... the former may or may not be desirable, but the latter is definitely not. Some studies have only found effects of “liking” punishment or punishers after repeated rounds. As such, I’m not convinced that the current methodology is optimal for testing this question, so it is hard to know what to

conclude from the null results of punishment. Personally, I suspect that the results probably are general – women are not attracted to punishers (unless they are in need of a protector) – but it would take more evidence to show this conclusively.

Response:

The reviewer questions the methodology and the interpretation. While we agree that there is always a problem of motivational interpretation, one-shot lab studies of punishment and compensation have shown reliable levels of punishment indicating that these paradigms *are* sufficient to produce punishment. Are men punishing to restore justice or because they like to punish? The type of evidence that the reviewer cites is based on studies that mainly only have an option to punish, and this may or may not be true in a restorative context. However, as the reviewer states the increase in punishment options happens after a number of rounds. Our design is actually more ecologically valid for restorative punishment. That is, in most real world contexts people are faced with a single restorative justice event at any one time to decide what to do, restorative justice does not usually unfold over time across repeat events (there maybe more than one event in a life time, but these will not be in quick succession for most people). As such, people are likely to face single acts and want to restore justice rather than to gain some positive reinforcements form the act of punishing solely. Thus our design is an ecologically valid representation of what would happen in the real world. Again – this can be highlighted briefly in the text.

Minor Issues:

In Experiment Two, when women were giving ratings of different types of men, presumably these were in counterbalanced order? If so, please specify. If not... that's problematic. Why not?

Response:

No. There could be potential order effect here. However, we were interested in the preferred choice and that was consistently compassion and second we do see an interacting by time frame that is consistent with the literature (in studies where counter balancing takes place). This is a limitation that we can acknowledge.

When standardised residuals are presented (p7, supplementary p4), What are these standardised residuals of? This is not clear.

Response:

The Standardised Residual is obtained by dividing the Cell Residual by the Square Root of the Expected Value for each cell.

Line 139: “punishment perceived as more attractive for ST than LT”... is this really attraction, or being less averse to punishers for ST? The ratings are below the mid-point of the scale. This warrants discussion, possible by giving somewhere (e.g., in Supplementary) what all points on the scale were labeled. Is a “4” someone of average desirability, such that scores below 4 mean that women actively dislike these guys?

Response:

The scales run from 1 not at all attractive to 7 very attractive. We do not include semantic labels on the other numbers to avoid possible interpretation bias around terms like average, somewhat a lot and so on. We agree the interpretation could be that women find these men less averse and as such finding them move towards the attractive end of the scale. Thus, we agree and can be more precise in our interpretation of the scale results.

*This also applies to the bit about “dark triad” (which always makes me cringe). In this case, are women actually *attracted* to these traits for ST, or are they just less averse to them in the ST (because unlike for LT they won’t experience those traits for long in a ST relationship). This is unclear.*

Response:

As with the point above we agree and can clarify the detail relating to the interpretation of the scale.

Where are the captions for Fig 1 & 2? Are the error bars SEM, SD, or 95% CI? Also, seem to be some typos in the figures (hard to tell, they’re so small)

Response:

There are only error bar for Figure 2 and they are 95%CI. This was stated in the figure legend in the main text in the submitted manuscript.

For the new control study, it would have been good to use the same number of participants, so that the bar graphs are comparable. I recommend either getting an equivalent number of participants in each condition, OR changing all graph axes from “Number of Participants” to “Proportion of Participants” (with raw numbers above). This holds for S1 and S2.

Response:

We can replot these as requested.

*Please note that the new control condition replicates the original control condition, but does not attempt to replicate the experimental condition. In other words, the *effect* was not replicated (or attempted to replicate). Thus, the authors should tone down claims about how the new control condition is a replication.*

Response:

We can tone this down.

P4 of Supplementary: last paragraph is hard to follow.

Response:

We can amend this.

Line 102: effect sizes of .02=small, .03=medium, 0.5=large... is this supposed to be .05?

Response:

We can amend this typo.

Thank you for considering this appeal, I really appreciate it. As a journal editor myself I know that appeals can be difficult for everyone, but after much consideration I felt that we have a case. I look forward to hearing from you and of course will abide whatever your final decision is.

Yours sincerely

Eamonn Ferguson

Additional Responses to the above Reviewer

It is more interesting to look at punishment, which is more novel, but it is also unclear that these laboratory situations would be enough to trigger a desire for a punisher. One-shot games are ambiguous about players' motives (hence "mixed-motive games"), so it is hard to tell the difference between someone who is punishing for a good reason ("restorative justice") and someone who is punishing because they like to hurt people... the former may or may not be desirable, but the latter is definitely not.

We have argued from the perspective of the theory of revealed altruism that a one shot 3PPC game is appropriate to identify men's preferences. We also assess moral outrage, trait empathic anger, as well as traits and emotions linked to compassion and it is only trait and emotion linked to compassion that predicts. Also the scores on these scale are within the ranges seen in the population and do not reflect low scores (empathy) and high scores (anger) that likely reflect traits (anti-social personality, psychopathy) associated with a simple desire to punish. Thus we feel our interpretation that punishment is a restorative mechanism in study one is a reasonable one

Line 139: "punishment perceived as more attractive for ST than LT"... is this really attraction, or being less averse to punishers for ST? The ratings are below the mid-point of the scale. This warrants discussion, possible by giving somewhere (e.g., in Supplementary) what all points on the scale were labeled. Is a "4" someone of average desirability, such that scores below 4 mean that women actively dislike these guys?

We have added analyses to the supplementary files to deal with this (p 24 in supplementary files) and revised the scaling in experiment 3 (1 = Unattractive, 4 = Neither Attractive nor Unattractive, 7 = Very Attractive).

For the new control study, it would have been good to use the same number of participants, so that the bar graphs are comparable. I recommend either getting an equivalent number of participants in each condition, OR changing all graph axes from “Number of Participants” to “Proportion of Participants” (with raw numbers above). This holds for S1 and S2.

We have replotted these figures in the main paper and supplementary files (p 22).

In Experiment Two, when women were giving ratings of different types of men, presumably these were in counterbalanced order? If so, please specify. If not... that’s problematic. Why not?

In experiment three we treated men viewed for a long-term or short-term relationship as a between subjects factor. The factor analytic results we now report for experiment 3 show that the ratings of the type of men (punishment preference etc.) are generally orthogonal.

Appendix B

The University of
Nottingham

UNITED KINGDOM · CHINA · MALAYSIA

Faculty of Science

School of Psychology

The University of Nottingham
University Park
Nottingham
NG7 2RD

t: +44 (0)115 951 5361

f: +44 (0)115 951 5324

e: psychology-enquiries@nottingham.ac.uk

www.nottingham.ac.uk/psychology

14th March 2019

Revision to: "Compassion Wins Out: Sexual Selection and Men and Women's Preferences for Punishment and Compensation") RSOS-181441

Dear Professor Hamilton,

Thank you for the opportunity to revise and resubmit this paper. I have responded to each of the reviewer's comments and these are detailed below. I have also highlighted in yellow where the relevant text has been changed.

Reviewer 1

Comments to the Author(s)

This paper explores the sexual selection of human pro-sociality using a version of a dictator game, in which a third party can compensate the victim of inequitable allocations, punish the dictator, do both, or do nothing. Study 1 shows that romantically-primed men more often chose to compensate or chose a combination of punishment and compensation compared to neutrally primed men. Study 2 shows that women rate men who chose compensation or compensation and punishment, in the decision situation above, more attractive compared to men who chose punishment or doing nothing. Participants were first asked to rate attractiveness when considering the person for a short-term relationship and afterwards when considering the person for a long-term relationship. Study 3 replicates study 2, randomly allocating participants to either the short-term or the long-term relationship conditions, as well as including ratings of indicators of cooperativeness and social status. It also investigates trait empathy and political ideology as correlates with preferences for compensation or punishment.

Issue 1

Page 7, line 136: The authors write ‘changing the choice architecture alters how respondents make choices’ citing Thaler and Sunstein (2009)’s ‘Nudge’. A further clarification is necessary how this relates to the introduction of third-party punishment to a game in which a third party can help/not help. Including punishment seems to be much more than a gentle ‘nudge’, changing the game’s strategies, possible payoffs and, depending on expected use of punishment, potentially the payoff-maximising strategy.

Response 1.

Munscher, Vetter and Scheuerle (2015) present a typology of nudges detailing how different choice architectures may influence behaviour. One component of this typology is the inclusion of extra choices to a choice set. Thus, changing the choice set of a 3PC game to include punishment should similarly alter the choice set and subsequent behavioural response. Specifically when a 3PC and 3PP games are played separately people both punish and compensate but exhibit significantly higher levels of compensation than punishment (Zhoa, Ferguson & Smiliie, 2018). When punishment and cooperation are possible, both are still exhibited, but levels of punishment are somewhat reduced (van Doorn et al., 2018, 2019a). Thus, we have added the following text [lines 101 – 122].

“Altering the choice architecture, is known to alter decision making [28, 50]. How would adding a punishment option to the standard *2-choice-altruism-architecture* alter the pattern of preferences? When people can help (compensate) or do nothing (a 3PC game) they choose to compensate and when people can do nothing or punish (a 3PP game) they choose to punish, with levels compensation significantly higher than punishment [23]. However, when people have to choose between punishment and compensation, levels of punishment are somewhat reduced [27, 51]. Thus, it appears that in a general context, when people have the option to compensate or punish, a preference away from punishment is observed. Thus, the full set of option (do nothing, punish, compensate or do both) should shift preferences towards compensation. How might this be further affected by a sexual selection context, where people are using their choices to display positive characteristics about themselves to attract a mate? In a simple *2-choce-altruistic-architecture* (e.g., ‘help or not-help’), altruism is signalled to a potential mate by helping [34]. In a *2-choce-altruistic punishment-architecture* of (e.g., ‘punish a transgressor or not’) trustworthiness is signalled to a potential mate by punishment [17]. However, in a *4-choice-altruism-and-punishment-architecture* it is less clear which choice best signals the potential mates best qualities. Thus, in a sexual selection context where altruistic punishment and victim compensation (altruism) are both possible, with each signalling very different yet desirable qualities [48], which is preferred? Thus, two questions remain, at present, unanswered. How would men choose to display their qualities as a mate if they could punish as well as help (altruism) following a transgression? How would women view male attractiveness based on men’s decision, following a transgression, when both options are available?”

- Munscher R, Vetter M, Scheuerle, T. A review and taxonomy of choice architecture techniques. *J Behav Dec Making*. 2016, **29**, 511-524. (Doi: 10.1002/bdn.1897)
- Van Doorn J, Zeenberg M, Breugelmans SM. (2019). An exploration of thirds parties' preferences for compensation over punishment: six experimental demonstrations. *They. Dec*. 85, 333-351. Doi. <https://doi.org/10.1007/s11238-018-9665-9>
- Van Doorn J, Zeenberg M, Breugelmans SM, Berger S, Okimoto TG. (2018). Prosocial Consequences of third-party anger. *Theory, Dec*. **84**, 858-599. Doi. 10.1007/s11238-017-9652-6
- Zhao K, Ferguson E, Smillie L. 2017 Politeness and Compassion Differentially Predict Adherence to Fairness Norms and Interventions to Norm Violation. *Sci Reps*. **7**, 3415. (doi: 10.1038/s41598-01702952).

Issue 2

Page 12, line 252: Did you mean the 'conditional information lottery' as described in Bardsley (2003), which is citation no. 68? I don't fully understand how this procedure was implemented here. A clarification would be very helpful. Please also describe what participants (in the role of player C) were told about players A and B. My main concern is: did the participants believe that their choices would have real payoff consequences for the other players?

Response 2.

Thank you for these observations, which are now clarified in the paper. In the Bardsley (2003) conditional information lottery (CIL) procedure participants experience a series of tasks (some real and some hypothetical) and know that they will be paid only based on their responses to the real tasks but do not know which are real or hypothetical and so have to respond to all tasks as if they are real. In our experiment there was one task (3PPC), so we selected a sub-set of participants. Evidence shows that the CIL method and selecting to pay a sub-set of participants rather than task are equally effective (Charness et al. 2016).

We have amended the text as follows [lines 238-242].

“To avoid deception and reduce transaction costs in such studies either a subset of tasks within an experiment [69] or sub-set of participants can be selected for payment. The evidence suggests that selecting participants, rather than tasks, has no discernible effects on the pattern of results [70-71]. Therefore, as we use a one-shot 3PPC game we selected to pay a subset of participants chosen at random.”

Charness G, Gneezy U, Halladay B. (2016). Experimental methods: Pay one or pay all *Journal of Economic Behavior & Organization* 131 (2016) 141–150 Doi [10.1016/j.jebo.2016.08.010](https://doi.org/10.1016/j.jebo.2016.08.010)

All participants in the role of player C know that their pay-off would be whatever they had left following their decisions to do-nothing, compensate, punish or do both. Player C was told that they were facing a hypothetical decisions on behalf of players A and B but that their decision would have a direct consequence of their own payoff. As such, they were

incentivized as they knew their decision had a real cost consequence for them. This methodology has been widely used to study 3rd party punishment and 3rd party compensation (Van de Vyver & Abrams, 2015; Lotz, Okimoto, Schlösser & Fetchenhauer, 2011; Leliveld, Dijk, Beest. 2012; Van Doorn, Zeeenberg, Breugelmans. 2019; Van Doorn, Zeeenberg, Breugelmans, Berger, Okimoto, 2018).

- Van de Vyver, J., & Abrams. D. (2015). Testing the prosocial effectiveness of the prototypical moral emotions: Elevation increases benevolent behaviors and outrage increases justice behaviors. *J Exp Soc Psychol.* **58**, 23-33. (Doi. 10.1016/j.jesp.2014.12.005).
- Leliveld MC, Dijk E, Beest I. 2012. Punishing and compensating others at your own expense: The role of empathic concern on reactions to distributive injustice. *Euro. J. Soc. Psychol.* **42**, 135-140. (doi:10.1002/ejsp.872)
- Lotz, S, Okimoto TG, Schlösser T, Fetchenhauer D. (2011). Punitive versus compensatory reactions to injustice: Emotional antecedents to third-party interventions. *J Exp Soc Psychol*, **47**, 477-480 (Doi.10.1016/j.jesp.2010.10.004)
- Lotz S, Baumert A, Schlösser T, Gresser F, Fetchenhauer D. (2011). Individual differences in third-party interventions: How justice sensitivity shapes altruistic punishment. *Negot. Conflict. Manag. Res.* **34**, 297-313. Doi. 0.1111/j.1750-4716.2011.00084.x
- Van Doorn J, Zeeenberg M, Breugelmans SM. (2019). An exploration of thirds parties' preferences for compensation over punishment: six experimental demonstrations. *Theory. Dec.* **85**, 333-351. Doi. <https://doi.org/10.1007/s11238-018-9665-9>
- Van Doorn J, Zeeenberg M, Breugelmans SM, Berger S, Okimoto TG. (2018). Prosocial Consequences of theird-party anger. *Theory, Dec.* **84**, 858-599. Doi. 10.1007/s11238-017-9652-6

We have amended the text in the paper as follows [lines 235-237].

“Participants were informed that the game between A and B was hypothetical but that they were playing for real money and the choice they made would constitute their final pay-off if they were selected to be paid.”

Issue 3.

Page 12, line 256: ‘8 participants would be randomly selected and paid based on their decision’. However, the instructions on page 2 of the Supplementary Files state: ‘At the end of the whole experiment 5 participants will be randomly selected and they will be paid based on their decision in the task below’. How many participants were actually paid?

Response 3.

Thank you for highlighting this. 8 participants were paid. We initially ran the ‘pure-control’ and ‘romantic-primes’ condition and told participants that 5 would be randomly selected. We later added the active-prime condition and in that we told participants that 3 would be selected at random. We have now added a footnote to this effect on page 11

“Overall eight participants were paid. For the no-prime and romantic-primes’ conditions, which were run first, participants were told that 5 would be randomly

selected. We later added the active-prime condition and in that we told participants that 3 would be selected at random.”

Issue 4

Page 14, line 299: ‘Women were presented with the same 3PCC game (individually or in a group setting: 44% vs 56%)’. Did you implement different conditions (i.e., individual choice vs a group choice)? An explanation would be very helpful.

Response 4.

Thank you for highlighting this. In all cases participants worked individually, the only difference was they were either on their own or in group (lecture setting). We have added the following text on lines 285-289

“Women were presented with the same 3PPC game as in Experiment One. In all cases women responded and made their choices individually. However, these individual choices were made in two context. Forty-four percent of the women made their individual responses on their own with no-one else present and 56% made their individual responses with others present in a lecture setting.”

Issue 5

Page 14, line 303: Participants rated the attractiveness ‘if he decided to spend his money on (1) punish player A, (2) compensate player B, (3) both punish and compensate or (4) keep the money’. What information was given to participants regarding the extent of compensation/punishment chosen by player C? For example, if player C would sacrifice the whole endowment to completely destroy player A’s payoff, this might be viewed as a very extreme action (and a potential signal of dark-triad traits), especially in case a less extreme punishment was available.

Response 5.

Thank you for this comment. We did not vary the amount spent on each of the strategies or the source of their money (property rights) or their relative wealth. Our aim was to establish the baseline effect, that is the effect of punishing, compensating, doing both or doing nothing. We acknowledge that this is an issue, but one we feel for further research. Indeed, in this context there is some evidence that the act itself may be as important as the amount. As such, we have added the following to the discussion [lines 655-661]

“With respect to the reaction of women we did not vary how much money men spent to punish, compensate, or do both, as in this series of experiments we were concerned primarily with establishing the basic effects of the behavioural act rather than their relative cost. Indeed, there is good reason to believe that the act is often as strong a signal as the cost of the act is [40]. Thus, we feel we have established the basic behavioural response and further research can explore how this is affected by the cost men are willing to incur to punish, compensate, or do both, as well as their relative wealth and the needs of players A and B.”

Issue 6

Please proofread and correct typos throughout the paper.

Response 6.

We have thoroughly proof read the paper.

Reviewer 2

This is an interesting set of studies. It's an interesting question: whether sexual selection has played a role in the evolution of altruism and punishment. I particularly like that the authors have tested both sides of the signaling – the effect of mating motives on compensation & punishment, and the effect of compensation & punishment on mate preferences. Both sides tell a similar story: mating motives have an effect on men's willingness to help others but not punish wrongdoers, and women prefer men who help others but not who punish wrongdoers. My main question is about the theoretical contribution and conceptual difference between altruism and compensation.

Issue 1

*Many authors have shown that altruism might have been selected for: men are more generous in mating situations, and women prefer generous men to neutral men. The authors show something similar, but by contrast, the authors stress that their results are about *compensation* and redistributive justice, rather than altruism per se. This is their main claim for novelty. Altruism and compensation are two very similar concepts: one individual pays a cost to confer benefits on another. It's not obvious why it's important to discriminate between them. Altruism occurs most when one individual has some need (e.g., empty belly), such that the help being conferred (e.g., one unit of food) increases the recipient's fitness more than it decreases the actor's fitness (i.e., $b > c$). The major difference between altruism and compensation is that in compensation, the need occurs because another individual has been unfair. By contrast, altruism is more general in that the need can come from any source (e.g., bad harvests), rather than just from exploitative conspecifics. Thus, compensation is a particular subset of altruism, where an actor confers benefits upon a recipient to alleviate a need caused by someone else's unfairness.*

If the authors wish to show that their results reflect compensation instead of altruism more generally, they will need to do both a) show that compensation is not a subset of altruism or generosity, and b) differentiate between them methodologically. In the current study, when participants compensate victims of unfairness, this is indistinguishable from generosity – one individual confers benefits upon another. The current methodology does not distinguish compensation from altruism (or generosity or whatever else one wishes to call it), and there is no way to retroactively do so. Thus, the results are just as consistent with "sexual selection for altruism" as they are with "sexual selection for compensation or redistributive justice". The former is relatively well-established, so it would not be new, but that is not a problem if RSOS publishes any study that is methodologically sound regardless of novelty (like PLoS ONE does). If the authors wish to claim that their study specifically looks at compensation (and NOT altruism), then their methods will have to do something to differentiate between these two motivations. Otherwise, they will have to rewrite the sections that stress the sexual selection of compensation, and be upfront that their results are equally consistent with (the well-established) sexual selection of altruism. So in my opinion, the authors have two choices for publication: 1) reframe the manuscript to reduce the stress on redistributive justice and

write it in terms of altruism; or 2) run a new study that differentiates between altruism and compensation. If novelty is not a deciding factor in publication at RSOS, then I would recommend the former (unless the authors really are most interesting in speaking about compensation). This reframing will need to be done in the introduction and the discussion (and abstract).

If by contrast the main contribution of this manuscript is about “resolving the tension between punishing and compensating”, then I’m less convinced that that’s an important question. I see the main contribution is in showing that punishment is not preferred in mating, but altruism is. That would be a useful addition to the literature.

Response 1

The reviewer makes an excellent point and on with which we agree and we now refer to compensation as a subset of altruism and site evidence showing that levels of 3rd party compensation and DG performance, as an index of altruism, are correlated. We have extensively rewritten the introduction [lines 35-136], discussion [lines 560-614, 634-638, 655-661] and abstract [line 15-21] as well as changed the title (“*To Help or Punish in the Face of Unfairness: Men and Women Prefer Altruism over Punishment in a Sexual Selection Context*”) to focus on the contribution – as suggested by the reviewer on “showing that punishment is not preferred in mating, but altruism is”.

Issue 2

On page 6 (lines 111-127), the authors attempt to argue that restorative justice is important because it relates to parenting skill. This is a stretch. I found this section unconvincing. Why is restorative justice particularly relevant here, beyond other forms of altruism? Why should punishment of unrelated third parties be related to demanding fairness among one’s own children? (Indeed, mothers might be the people who do the most enforcement of fairness among their own children, but do the least third-party punishing.) To be honest, this sounded like post-hoc hypothesizing on why restorative justice is particularly relevant above and beyond other forms of altruism. As such, this section requires either significant strengthening or elimination.

Response 2

We agree and have eliminated this section.

Issue 3

The analyses in section 3.3.5 are very hard to understand. This needs to be clarified. Similarly, Table 2 is hard to understand. What are the columns? How that trait loads on preference (for male type)... as measured by what? It seems a bit circular, that something about compensate and punishment would load heavily on compensate and punishment. If these are factors, then don't label them as male types. At the very least, this Table needs to be much better explained.

Tables 3 and 4 could be clarified a bit as well.

Response 3

We reworked this section. We have used a confirmatory rather than exploratory factor analytic approach to clarify why this is important as there are alternative potential models to explain the covariation between these ratings. As such, we have added the following text [lines 458-484]

“3.3.4. Covariance of Ratings

Based on Miller and Todd's [4] lens-model we would expect that the four ratings (attractiveness, compassion, fairness and strength) to positively covary with each other. However, there are two possible ways they can covary. The first is *generic* and regardless of context (compensation, punishment, doing both or doing nothing), indicating judgements of phenotypes are context independent. That is, people's four ratings of compassion are all associated with each other regardless of the targets actions (e.g., punish or compensate), with raters having a preference to rate compassion in a particular way. With same true of fairness, strength and attractiveness (Model 1). However, context may matter and people make ratings of the target taking into the context in which the target is acting? In which case ratings of attractiveness, compassion, fairness and strength should all covary within each context (e.g., punish or compensate) (Model 2). To explore these possibilities we use Confirmatory Factor Analysis (CFAs) to specific these two models. The models were specified in *MPlus* 8.1 [78] using a weighted diagonal least squares (with means and variance adjustments) estimator. Model 1 had four factors with factor-1 comprising all 4 compassion ratings regardless of the preference, factor-2 all 4 strength ratings, factor-3 all 4 fairness ratings and factor-4 all 4 attractiveness rating. Model 2 also specified 4 factors but this time the factors represented ratings of compassion, strength, fairness and attractiveness within each preferences. Thus factor-1 represented these 4 rating when men 'both compensate and punish', factor-2 when men 'do-nothing', factor-3 when men 'compensate' and factor-4 when men 'punish'. Fit was assessed in terms of the Tucker-Lewis Index (TLI: which should be greater than .95), the Comparatives Fit Index (CFI: which should be greater than .95) and the root mean square error of approximation (RMSEA, which should be .06 or less and non-significant) [79]. The fit for model 1 was poor ($TLI = .48$, $CFI = .58$, $RMSEA = .31$ $RMSEA$ p value = .000). However, the fit for model 2 was excellent ($TLI = .98$, $CFI = .98$, $RMSEA = .06$ $RMSEA$ p value = .112). The factor loadings for Model 2 are shown in Table 2. Thus, judgements of overall mater quality are context dependent.”

We have now clarified table 2, but we have also added the following to the table legend.
*“Note. Columns refer to loading of each judgment on its specified factor. *** $p < .001$ (N = 159).”*

Issue 4.

It’s a bit ambitious to say that the authors are aiming for a power of 80% to detect Cohen’s $d=0.8$ (large effect size). If the authors wish to say that they had sufficient power to detect an effect of punishment, then they should have done the power analysis with a more realistic effect size (small-medium of 0.35 or medium of 0.5). At this point, now that the data are collected, the authors should do some sort of sensitivity analysis or post hoc power analysis: how big would the effect size have to be to have an 80% chance of detecting it with their sample size? Or something similar, to show that the lack of significance is not just due to low power.

Response 4

We have added text to who the Ns required to have an 80% power with a of .05 given the effect sizes in the paper [line 357-360].

“Given the effect sizes between the ‘romantic prime’ and ‘combined control’ conditions, 19 participants are need to achieve a power of .80 with an α of .05 for variation in displays of compensation, 19 for ‘doing nothing’, 30 for punishment and 48 for ‘both compensate and punish’.”

Issue 5

In Study 1, how much time was there between the prime and the dependent measure? Is it likely that the prime would last that long?

Response 5

The 3PPC game followed 5 minutes after the primes [lines 268-269].

Issue 6

Is there an overall negative effect of punishment in Study 3?

Response 6

We show that overall that punishment results in lower rating across the board (compassion, fairness, strength and attractiveness) compared to ‘compensation’ and ‘both compensation and punishment’ but is not significantly different from ‘doing nothing’. We feel that if there were an overall negative effect of punishment then it should result in lower judgements by women than doing nothing. As such, I have not altered the text in this case. But if we have misunderstood your point, our apologies, and we can also make further revision if necessary.

Issue 7

There are multiple typos, including in the Figure 1 caption.

Response 7.

We have fully proof read the paper and hopefully the number of typos is minimized.

I hope that we have responded fully to all the reviewers comments and that the paper is now more suitable to be considered for publication in *Royal Society Open Science*.

I look forward to hearing from you.

Yours sincerely,

Eamonn Ferguson

Appendix C

The University of
Nottingham

UNITED KINGDOM · CHINA · MALAYSIA

Faculty of Science

School of Psychology

The University of Nottingham

University Park

Nottingham

NG7 2RD

t: +44 (0)115 951 5361

f: +44 (0)115 951 5324

e: psychology-enquiries@nottingham.ac.uk

www.nottingham.ac.uk/psychology

17th May 2019

Revision to: "RSOS-181441.R1 entitled "To Help or Punish in the Face of Unfairness: Men and Women Prefer Altruism over Punishment in a Sexual Selection Context"

Dear Professor Hamilton,

Thank you for the opportunity to revise and resubmit this paper and the supportive and constructive comments of the reviewer. I have responded to your comments and the reviewer's comments in detailed below.

Editor's Comments

As the original reviewers of your RSOS-181441 were unavailable, a new referee has commented on your work. They see merit in the publication; however, they also point out a number of problems with the way it is presented (among other matters). As the data is interesting, we will allow one further revision but must warn that if the reviewers and editors are not happy with the clarity of the writing and links to theories (see comments below), the paper will be rejected with no further option to resubmit.

Response.

Thank you for the opportunity to revise the paper one more time, it is appreciated. I have extensively re-written the introduction to take on board the reviewer's comments and used consistent terminology drawing on the frameworks suggested by the reviewer (as well as other related ones of Bshary and Bergmüller (2008) and Pizzari and Gardner (2012)) [lines 60 to 152]. I have summarized this terminology and links to theory in Table 1 as well as in an extensively re-written introduction [lines 60 to 152 with addition subheads and sections 1.1.1 (lines 73-114) and 1.1.2 (line 115 to 152) and 1.2 (lines 155 to 201)]. I have also extended the analyses of sexual selection theory [lines 41 to 59 and lines 657 to 705, 706 to 731 as well as 751 to 764] and restructured the narrative flow of the introduction, with the methods [lines 155 to 201] and measures [line 202 to 260] to be used are detailed more clearly in the introduction and the methods section should follow logically. We have also amended the title to fit with the theoretical conceptualization of the paper on *cooperation* and *self-serving mutually beneficial behaviour*.

Reviewer: 3

I think that the authors have interesting data here, and I like that they looked at both the signaling side (here, male behavior in a simulated mating context) and the signal receiver side (here, female preference for men as romantic partners, contingent on male behavior). I want to see these data published, but I find it very difficult to interpret this work as currently written so I recommend revision to enhance clarity.

Response.

We thank the reviewer for their supportive comments.

Two problems, perhaps related, jump out at me immediately: inconsistent terminology, and unclear connections to functional theories about mating psychology and cooperation more generally. I further describe these two major problems, then note some other issues.

Major problem 1: Definitions of key terms

The central concepts of the paper are not clearly defined, and key terms seem to be used in different ways in different sections. Altruism is introduced by way of example rather than definition. Altruism at first seems to describe behavior -- helping, generosity, etc. Then in line 65 altruism seems to be an individual trait that is signaled by helping behavior. The usage goes back and forth after this.

Furthermore, punishment is introduced as "altruistic" and called a "key mechanism for the evolution and survival of altruism," but then punishment is contrasted with "altruism" for the rest of the paper. This is extremely confusing. For coordinating terminology, especially for those of us trying to link human psychology to biological theory, I always recommend West et al 2007 (<https://doi.org/10.1111%2Fj.1420-9101.2006.01258.x>). Other semantic schemes could also work, as long as terms are used in a clear, consistent way.

Response 1

We thank the reviewer for this. We have now completely re-worked the introduction to focus on cooperation and not altruism and made the following major changes. First, we have used the schematic frame work of West and colleagues (as recommended) as well as Bshary and Bergmüller (2008) and Pizzari and Gardner (2012) [lines 60 to 201] to define the focus of the previous literature more clearly and what is missing from it [lines 73 to 114], why we need to study punishment [lines 115 to 152] and to define the game we use, the 3rd party punishment and compensation (3PPC) game, in terms of the behaviours assessed [lines 155 to 201]. This is also all summarised in a new Table 1.

We use this to show that the existing literature on the sexual selection of cooperation focus on self-serving mutually-beneficial helping and mutual cooperation [lines 73 to 152].

Having defined the behaviours we are assessing in the 3PPC game, we are able to use this to explain some of the finding more precisely, in particular suggest possibilities why women's ratings of strength are greatest for compensation [lines 682 to 692 & 736 to 742].

We have also amended the title to fit with the theoretical conceptualization of the paper on *cooperation* and *self-serving mutually beneficial behaviour*.

Bshary R, Bergmüller, R. 2008. Distinguishing four fundamental approaches to the evolution of helping. *J. Evol. Biol.*, **21**, 405-420. (Doi. 10.1111/j.1420-9101-2007.01482.x)

Pizzari T, Gardner A. (2012) The sociobiology of sex: inclusive fitness consequences of inter-sexual interactions. *Philos. Trans. R. Soc. Lond. B. Biol. Sci.* 367, 2314–2323. (Doi: [10.1098/rstb.2011.0281](https://doi.org/10.1098/rstb.2011.0281))

West SA, Griffin ASA, Gardner A. (2007) Social semantics: altruism, cooperation, mutualism, strong reciprocity and group selection. *J Evol. Biol.* 20, 415-432. (Doi. doi.org/10.1111/j.1420-9101.2006.01258.x)

West SA, Mouden CE, Gardner A. 2011. Sixteen common misconceptions about the evolution of cooperation in humans. *Evol. Hum. Behav.* **32**, 231-262 (Doi. 10.1016/j.evolhumbehav.2010.08.01))

Major problem 2: Tenuous links to evolutionary theories

In the opening paragraph, the authors assert that sexual selection on altruistic behavior must involve female preference for male behavior; this ignores the obvious logical possibility of selection by male preference for female behavior. It is fine for the authors to only investigate one direction of sexual selection; it is not fine to claim the other direction does not exist. Doing so in the opening paragraph sets an unfortunate tone that the connection to relevant work might be too superficial. Indeed, the authors cite a very limited subset of relevant literature in support of their errant claim. The authors go on to cite other relevant literature, but haven't synthesized its relevance to the current point. I urge the authors to engage more deeply with recent theory and evidence. And, again, the confusion about terminology described above, makes it hard to connect the present work to ultimate theories about the evolution of (certain types of) cooperation, versus proximate theories about mechanisms involved in cooperative behavior (e.g., empathy), etc.

Response 2

Again we thank the reviewer for this very helpful comment. As per the comments above we have extended the discussion and explanation of sexual selection at the start of the introduction [lines 41 to 59] and in the discussion [657 to 705] to highlight the complex nature of sexual selection and that we just focus on female choice. We have also added in a section on links between social selection and sexual selection and how our results talk to this debate [lines 706 to 731]. We hope that this make the justification of the study clearer. We have also made the ultimate-proximal distinction clearer throughout [lines 38 to 41, 297-260].

Clutton-Brock T. (2007) Sexual Selection in Males and Females. *Science.* **318**, 1882-1885 (Doi: 10.1126/science.1133311)

Crook, J. H. 1972 Sexual selection, dimorphism, and social organization in the primates. In *Sexual selection and the descent of man (1871–1971)* (ed. B. G. Campbell), pp. 231–281. Chicago, IL: Aldine

Diaz-Munoz SL, VuVal EH, Krakauer AH, Lacey EA. 2014. Cooperating to compete: altruism, sexual selection and causes of male reproductive cooperation. *Anim. Behav.* **88**, 67–78. (Doi. 10.1016/j.anbehav.2013.11.008)

Kirkpatrick M, Hall DW. 2004. Sexual selection and sex linkage. *Evol.* **58**. 683–91 (Doi. 10.1111/j.0014-3820.2004.tb00401.x)

Kokko H, Johnstone RA. 2002. Why is mutual mate choice not the norm? Operational sex ratios, sex roles and the evolution of sexually dimorphic and monomorphic signalling. *Philos. Trans. R. Soc. Lond. Ser. B.* **357**, 319–30 (Doi. 10.1098/rstb.2001.0926)

Kuijper B, Pen I, Weissing FJ. (2012). A Guide to Sexual Selection Theory. *Ann. Rev. Ecol. Evol. Syst.* **43**, 287–311 (Doi. 10.1146.annerv-ecolsys-110411-160245)

Luoto S. (2019). An Updated Theoretical Framework for Human Sexual Selection: from Ecology, Genetics, and Life History to Extended Phenotypes. *Adapt. Hum. Behav. Physiol.* **5**, 48–102 (doi.org/10.1007/s40750-018-0103-6).

Lyon BE, Montgomerie R. 2012. Sexual selection is a form of social selection. *Philos. Trans. R. Soc. Lond. B. Biol. Sci.* **367**, 2266–2273. (Doi: 10.1098/rstb.2012.0012).

Miller GF, Todd M. 1998. Mate choice turns cognitive. *TICS*, **5**, 190–198 (Doi 10.1016/S1364-6613(98)01169-3)

Miller GF. 2007. Sexual selection for moral virtues. *Q. J. Bio.* **82**, 97–125. (doi:10.1086/517857)

Ness RM. 2007. Runaway social selection for displays of partner value and altruism. *Biol Theor.* **2**, 143–155. (Doi. 10.1162/biot.2007.2.2.143)

Roswell KA. 2011. Intrasexual competition in females: evidence for sexual selection? *Behav. Ecol.* **22**, 1131–1140 (Doi. 10.1093/beheco/arr106)

Roughgarden J. 2012. RThe social selection alternative to sexual selection. *Philos. Trans. R. Soc. Lond. B. Biol. Sci.* **367**, 2294–2303. (Doi: 10.1098/rstb.2011.0282)

Stockley P, Campbell A. 2013. Female competition and aggression: interdisciplinary perspectives. *Phil. Trans. R. Soc. B.* **368**, 20130073. (Doi. 10.1098/rstb.2013.0073)

Other notes:

I see that data and materials are public. I do not see analysis code shared anywhere; apologies if I'm overlooking this.

Response 3

I have added this to supplementary files in the Supplementary File in Section E. I have also added extra data file to the publically available data to save others from needing to restructure that data files for the different analyses reported

Generally, I don't understand why many of these measures and methods are being used. The introduction doesn't adequately prepare me when the methods are presented.

Response 4

I have re-worked the introduction to highlight the methods [lines 155 to 201] and measures [line 202 to 260] to be used more clearly and as such the methods section should follow logically.

"Attractiveness" seems intended to mean physical appearance, but unclear whether participants might interpret it more broadly. Same with "strength," which seems intended to mean physical muscular strength, but could be interpreted to include "strength of character" or the like.

Response 5

This is a good point. I have added a comment on this in the limitation section [lines 786 to 799], to acknowledge this and suggest future research possibilities.

Study 1 -- So participants are basically paying real money to send a signal to experimenter? Do we need to be concerned about personal characteristics of the researcher(s) and how the participants responded to the real human interaction? Did the same experimenter administer all sessions?

Response 6

The same experimenter administered all the condition in experiment one. We tried to reduce any experimenter effects by having financial decision made anonymously. The experimenter was not present when financial decisions were made (or any other aspects of the study), these were placed in sealed envelope and in a sealed container by the participant. The experimenter passed these on for payment. We have tried to emphasize this more clearly in the methods [321; 358 to 361].

Study 2 -- Why is short/long the right dichotomy here? It seems more like serious/casual might be more appropriate. Is public/private an experimental variable? Is this an "experiment"?

Response 7

We have used the short-term vs long-term distinction for two reasons [lines 215 to 228]. First, it is the distinction used in all the studies on the sexual selection of cooperation, so we can be consistent with that literature. Second, the sexual strategy theory of Buss suggest that of all the distinction is important as men and women look for different characteristics for short-term vs long-term relationships. In this study some participant completed the measure son their own and some in a group. In all cases their responses were completed in private – so this was not an experimental variable. We show in the supplementary files that this did not affect the results. We agree this is not technically and experiment and there is no randomization to conditions and participants are not in non-randomized to groups (a quasi-experiment). As such we now refer to all 3 reported ‘experiments’ as studies.

Results 3.3.4 and 3.3.5 are very hard to follow.

Response 8

We have re-written sections 3.3.4 [lines 549 to 579 and 3.3.5 [lines 580 to 642] to explain why the analyses are need and provide a clearer summary of the fining

I hope that we have responded fully to all the reviewers comments and that the paper is now more suitable to be considered for publication in *Royal Society Open Science*.

I look forward to hearing from you.

Yours sincerely,

Eamonn Ferguson

Appendix D

The University of
Nottingham

UNITED KINGDOM • CHINA • MALAYSIA

Faculty of Science

School of Psychology

The University of Nottingham

University Park

Nottingham

NG7 2RD

t: +44 (0)115 951 5361

f: +44 (0)115 951 5324

e: psychology-enquiries@nottingham.ac.uk

www.nottingham.ac.uk/psychology

11th July 2019

Revision to: "RSOS-181441.R2 entitled "To Help or Punish in the Face of Unfairness: Men and Women Prefer Altruism over Punishment in a Sexual Selection Context"

Dear Professor Hamilton,

Thank you for the opportunity to revise and resubmit this paper and the supportive and constructive comments of the reviewer. I have responded to the reviewer's comments as detailed below.

Reviewer: 3

As I said before, I like a lot of what the authors did here. The 4-choice design is an excellent extension of prior work, and looking at both the signaler and receiver side is great. The authors have made much clearer connections to the relevant literature in this version, which is much appreciated.

And, as before, I want to see these data published, but the manuscript in its present form is not yet suitable for publication. I list some concerns below. I believe that they can be addressed with minor revision.

RESPONSE 1

We thank the review for their supportive comments.

1.

*First, this work is based on ~350 British undergraduate participants. I appreciate that Line 747-749 acknowledges that mating-related preferences might vary with other aspects of social systems, and that Section 4.4 discusses cross-cultural considerations. But I am concerned that these crucial caveats are undermined with inflated claims, e.g. "We provide some initial evidence that sexual selection, based on female choice, favours compensation over punishment" or when the abstract simply says "We test which is preferred by men and chosen by women," etc. I think the authors have shown what sexual selection *can* favour, and they've shown what is preferred and chosen *by undergrads at their University*. There are other places where the language is similarly excessive in my judgment.*

RESPONSE 1

We have amended and toned down these claims in a manner consistent with the reviewer's comments. In particular, we have changed the text in the abstract and lines 766 to 777 and throughout the text.

2.

Next, and less substantively, proofreading is badly needed. I noticed lots of errors of spelling, grammar, punctuation, etc. I'm sympathetic, being prone to such errors myself. Here are some:

- *"with respect [to]" (lines 80, 126)*
- *"High levels [of] cooperation" (line 125)*
- *"self-severing" lines 138, 180*
- *"Empathy is ... a mechanism[s]" lines 242-3*
- *"in two context[s]" line 379*
- *"overall ratings of the targets attractiveness" line 557*

RESPONSE 2

Thank you for this. These have been corrected and the paper fully proof read.

3.

I previously noted that there was no real person being punished/compensated, and suggested that the only way paying to alter the payoffs of a fictional character wasn't a total waste is if participants were trying to signal to the experimenter, in which case maybe we need to think about the participant-experimenter dynamic. I'm fine with the authors' response regarding anonymity, but I do need to raise another concern with a fake Player A: From the perspective of the female rater, why shouldn't we expect her to think that men who just give away their money to punish or compensate non-existent characters aren't the best partners? Shouldn't "do nothing" be the most attractive option, at least for a subset of women who see through the fiction? (Would presumably apply in the broader social selection framework too – do we want an ally who can't distinguish reality from fiction?) I don't think that another study with real stakes is required, but this concern should be discussed. (And it isn't totally clear that the women knew that the men were punishing/compensating a fake Player A. If they didn't know, this concern disappears, although then we might need to discuss the use of deception, unless the hypothetical-ness was clear.)

RESPONSE 3

In study 1 the male participant (player C) responds to a scenario where player A unfairly treats player B and can then choose to do nothing, punish, compensate or do both. Their choices resulted in real financial consequences for the participant (player C), but given the nature of the scenario there was no real consequences for players A and B. The question is whether this fundamentally alters the choices made by Player C compared to responses that would actual help or harm players B and A respectively. We argue that this does not alter the nature of the decision. This argument is detailed on line 322 to 330 as follows.

“Given that players A and B were hypothetical, then the decision made by the participant, player C, will affect their own pay-off but not players A and B. Thus, the participants decision signals their *intent* to

punish or compensate. If participants were only concerned about the direct effects of their actions on players A and B, rather than what their actions signal about them, they would keep the money and do nothing. As there is evidence that the intentions are important signals about reputation [69, 89] as well as self-signals about the person's own motivations [90], we feel that participant's choices will be an indication of their preferences. Furthermore, there is evidence that preferences in hypothetical games are similar to those in real game [91-93]."

In studies 2 and 3 the women were presented with a vignette concerning how men choose to respond (e.g., punish, compensate etc.) when faced with an unfair allocation from player A towards player B in a 3PPC game. Thus, they were responding to this as if it were a real scenario they have been informed about. This vignette methodology is commonly used to study female preference in a sexual selection context. We make this more explicit on lines 383 to 387 and lines 414 to 417.

4.

"Women preferred strength for long-term relationships... Women preferred attractive men for short-term relationships" – I don't understand what these claims are based on. Women rated hypothetical men's strength and attractiveness as a function of the men's decision and the women's relationship context. So we can know which male actions women prefer, but how do we know which traits they prefer in different contexts? Aren't the traits entirely based on the actions?

RESPONSE 4

We manipulated relationship length (within subjects in study 2 and between subjects in study 3). The results for the effect of relationship length in study 3 (lines 546 to 553) showed that women rated men overall as more attractive when considering a short-term relationships and as strong when considering a long-term relationship. In study 2 (504-507) women also rated men as more attractive when considering short-term relationships.

5.

In general, since 3.3.4 suggests that all of the ratings load to some "mate quality" factor, and since "strength" seems to have a fuzzy meaning as deployed here, it is worth trying to parse the meanings of each individual rating word?

But if we want to do that, it seems to me that punishment could signal literal physical strength when the mode of punishment involves physical action (e.g., detaining and/or assaulting the perpetrator), whereas the mode of (fake) punishment here was economic. The first part in the general discussion about "strength" (line 682-92) was entirely based on a metaphorical notion of the word – ignoring the primary (somatic) definition of the word seems odd.

RESPONSE 5

We tried to detail other interpretations of strength in the discussion as a way to stimulate future research [lines 793-803]. However, given that ratings of strength load on a factor with ratings of compassion, fairness and attractiveness suggest more that strength in this context is

linked to “strength of character” rather than physical strength. We have added clarification about this in the discussion as follows on lines 793 to 803.

“Similar issues arise concerning the assessment of strength that could represent either physical strength or ‘strength of character’. Work in the area of ‘strength of character’ has identified the presence of an ‘other-directed’ character strength that contains kindness, modesty and bravery [121]. Consistent with this the results from the factor analytic models reported in this paper show that ratings of strength loaded with ratings of fairness, compassion and attractiveness. Thus, we feel that strength in this context is about character virtues rather than physical strength.”

6.

Table 1 and related text: Could punishment have long-term reputation costs, e.g. a reputation for violence or cruelty? (And, as mentioned previously, could punishment or compensation of fictional characters have reputations costs?)

RESPONSE 6

We have added the idea that punishment may have long-term reputation costs to table 1 and in the text [lines 127-129 and 168-170].

We refer the referee to our response to point 2 above why we feel the fictional context of study 1 would lead to real preferences, and that the vignettes of studies 2 and 3 represent real information presented to the female participants.

7.

From footnote 1, it sounds like the Study 1 “active control” condition participants were run months after the others. I’d suggest stating that explicitly rather than leaving it implicit.

RESPONSE 7

We have made this explicit.

8.

I don’t object to the content of section 4.2, but it seems unnecessary in this paper, given that the methods were clearly aimed at mating psychology, rather than a more general cooperative psychology. Acknowledging that sexual selection can be seen as a subset of social selection (or perhaps other interpretations) and crediting important work on this topic could be accomplished far more efficiently, probably in the Intro.

RESPONSE 8

We feel that the exposition of the sexual vs social selection debate was important to the focus of the paper but we felt it was more of a discussion point and, as such, we have left it in the discussion. But if the referee still feels that it needs to be moved we are happy to try and accommodate such a request.

9.

I suspect that the paper would read more smoothly if a lot of the technical details of 3.3.4 and maybe 3.3.5 were relegated to Supplement, with only summaries of that stuff in the main text.

RESPONSE 9

We have moved the technical detail to the supplementary files (supplementary file E) as suggested.

10.

Figure text is difficult to read. Could be the journal system's auto-PDF distorting it?

RESPONSE 10

We hope this will be resolved in the final published version.

We hope that these changes make the paper now suitable for publication in RSOS.

Yours sincerely,

Eamonn Ferguson